# Rich spatio-temporal stimulus dynamics unveil sensory specialization in cortical area S2

Matías A. Goldin [1], Evan R. Harrell [1], Luc Estebanez [1] & Daniel E. Shulz[1]

Tactile perception in rodents depends on simultaneous, multi-whisker contacts with objects. Although it is known that neurons in secondary somatosensory cortex (wS2) respond to individual deflections of many whiskers, wS2's precise function remains unknown. The convergence of information from multiple whiskers into wS2 neurons suggests that they are good candidates for integrating multi-whisker information. Here, we apply stimulation patterns with rich dynamics simultaneously to 24 macro-vibrissae of rats while recording large populations of single neurons. Varying inter-whisker correlations without changing single whisker statistics, we observe pronounced supra-linear multi-whisker integration. Using novel analysis methods, we show that continuous multi-whisker movements contribute to the firing of wS2 neurons over long temporal windows, facilitating spatio-temporal integration. In contrast, primary cortex (wS1) neurons encode fine features of whisker movements on precise temporal scales. These results provide the first description of wS2's representation during multi-whisker stimulation and outline its specialized role in parallel to wS1 tactile processing.

[1] Unité de Neurosciences, Information et Complexité, UNIC-FRE3693, Centre National de la Recherche Scientifique, Gif sur Yvette 91198, France. These authors contributed equally: Matías A. Goldin, Evan R. Harrell. Correspondence and requests for materials should be addressed to D.E.S. (email: shulz@unic.cnrs-gif.fr)

The whisker system in rodents supports localization and identification of objects in the environment through precisely controlled spatio-temporal patterns of contact with the mystacial vibrissae. At the neuronal level, representations of vibrissal movements and contacts have been extensively studied in the primary somatosensory cortex (wS1) in different laminae, cell types, and topographical locations[1,2]. Much less attention has been given to the whisker secondary somatosensory area (wS2), which like wS1 contains a somatotopic map of the mystacial vibrissae[3–5], but lacks the detailed topographical structure found in the barrel cortex. In narcotized rats, single neurons in wS2 yield rapidly adapting responses to individual deflections of many whiskers. Each whisker eliciting a response can do so with a different latency[6]. The fastest wS2 responses are in the same time range as the low-latency responses found in wS1 neurons[7], which suggests that at least some of the responses are driven by direct thalamic input and do not depend on wS1 activation.

A wealth of anatomical data exists on the connectivity and hierarchical organization of the somatosensory system in rats. Many anatomical tracing studies suggest that based on the thalamic input structure and the nature of their reciprocal connectivity, wS2 and wS1 can be thought of as equally placed in the somatosensory hierarchy[8–11]. Direct thalamic input to wS2 comes through both the extralemniscal and paralemniscal pathways[12], while corticocortical projections from S1 to S2 originate in layers 2, 3, and 5a of S1 and terminate in extragranular layers of S2. The reciprocal connections from S2 to S1 follow the same connection pattern[9]. In the mouse, infragranular cells in wS1 receive more numerous connections from infragranular wS2 cells, and likewise supragranular wS1 cells receive more connections from supragranular wS2 cells[11]. Although there is still some controversy, these data suggest that sensory processing at the level of wS2 and wS1 in rodents is done in an interdependent, parallelized manner.

While there is anatomical evidence for parallel processing of somatosensory information in wS2 and wS1, recent perceptual studies in mice have highlighted wS2's role in the choice-related, top-down flow of information[13,14]. Although choice-related activity in wS2 is more predominant than it is in wS1, it is unknown whether the single whisker periodic deflection used in these studies adequately engages wS2's sensory representation. With such a stimulus, there is no global or multi-whisker component, which could be an important factor in delineating wS2's sensory function[6,15]. With this in mind, we set out to assess wS2's sensory representation during multi-whisker stimulation. To this end, the first important question to ask is what patterns of whisker movement are salient for wS2 neurons, and how these whisker movement patterns differ from those represented in wS1. Accordingly, this work aims to provide a detailed characterization of the responses of large populations of single units recorded in wS2 during multi-whisker stimulation.

Using a custom-built multi-whisker stimulation matrix[16], we applied different types of Gaussian white noise to the caudal 24 vibrissae in anesthetized rats. We introduce novel analysis methods to extract spatio-temporal stimulus dependencies during continuous multi-whisker stimulation. We describe wS2 and wS1 using this new dynamic receptive field method and reveal specialization between the two regions. The stimulus dependencies in wS2 are extended in space and time, which facilitates the computation of the global statistics of the tactile scene, while wS1 contains a more pronounced representation of spatially local and temporally precise stimulus features. Correlated whisker movements produce larger supra-linearities in the responses of wS2 neurons than seen in wS1. These specialized and complementary cortical representations can support the large range of tactile discrimination abilities that rats are known to possess and have long been hypothesized to exist along the somatosensory pathway.

## Results

**Temporal integration is longer in wS2.** While it has been reported that wS2 neurons respond to individual deflections of many single whiskers, some with latencies comparable to wS1 cells[6], it is still unknown how they respond during continuous multi-whisker stimulation. To carry out this analysis in wS2, we recorded the activity of 1157 single units during globally correlated Gaussian white noise whisker movements along the rostro-caudal axis (Fig. 1a). To obtain sufficient spiking activity, all recorded neurons from wS2 were in granular or infragranular layers (layers 4, 5, and 6). From the activity observed during these correlated stimulations, we applied spike-triggered covariance (STC) analysis. Briefly, this method entails tabulating the whisker movement shapes preceding each spike from a single unit, computing the covariance matrix of these whisker movements, and using eigenvector decomposition of this matrix to find the whisker movements, or filters, responsible for eliciting the most spikes (See Methods for details). Using this approach, we obtained 411 significant linear filters from 204 of the neurons recorded in wS2. To compare, we applied the same analysis to 1038 neurons recorded previously in wS1[17], which yielded 1320 significant linear filters from 567 responsive neurons. Thus, there is a major difference in the number of cells yielding significant filters in the two regions (204/1156 in wS2 vs. 567/1038 in wS1). This observation could be due to a real difference between the whisker movements coded in the two regions (with wS1 more strongly activated by features contained in globally correlated white noise) or reflect a difference in the signal-to-noise-ratio (SNR) of the responses in the two regions (where less spikes/stimulus in wS2 results in lower filter yields).

Focusing for now on the cells giving filters, single wS2 neurons showed as many as four significant filters. In both wS2 and wS1, we performed a PCA on all the significant filters and found that the two predominant dimensions explain 88.96 and 86.77% of the variance in the filter shapes of the respective populations (Fig. 1b, c). This implies that the populations of filters in both wS2 and wS1 can be well-represented by a two-dimensional "phase" space (Fig. 1d, e). Most of the individual significant filters lie very close to the unit circle when projected into this phase space, meaning that they can be described as a linear combination of two representative filters. The top two significant filters for each neuron (filled circles in Fig. 1d, e) are particularly well-described in these two dimensions. This indicates that across the populations of neurons in wS2 and wS1 and within the statistics of our globally correlated stimulation, these relevant phase spaces provide a much-reduced, concise description of the stimulus dependence of the firing in each cortical area.

To easily compare the phase spaces of wS2 and wS1, we phase-aligned and overlaid them, as shown in Fig. 1f. Although the shapes appear similar, wS1 filters have more energy close to the spike time and wS2 filters have larger stimulus dependencies at longer temporal delays. To assess the significance of this observation, we computed the temporal envelope of the subspace filter energies for wS2 and wS1 (Fig. 1g, h) and carried out a permutation test by shuffling the cell-to-region mappings. We found that in wS2 both the half-width of the filter energy distribution and the long-latency filter energy (time at which one half of the energy is to the left) are significantly larger (by ~ 4–6 ms) than wS1 (Fig. 1h, $p < 0.001$, permutation test). This highlights that the temporal integration windows in the two regions are significantly different. Even if both cortical areas contain low-latency sensory responses, wS2 integrates whisker movements over a longer time window than wS1.

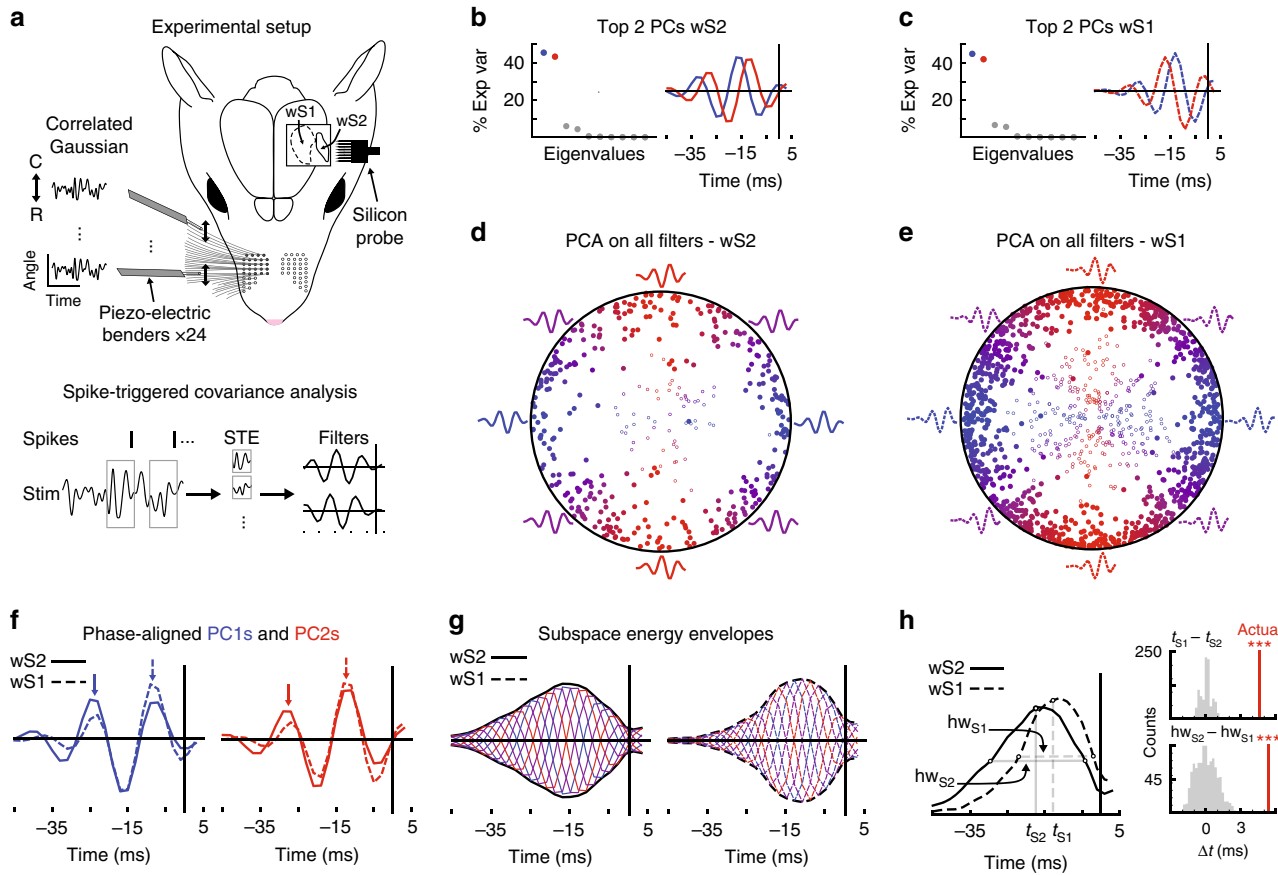

**Fig. 1** Extended temporal integration in wS2 during correlated Gaussian white noise stimulation **a** Recording setup. Top: The right 24 caudal macrovibrissae of isoflurane-anesthetized rats are trimmed and placed inside plastic tips attached to piezo-electric benders. Only two stimulators are shown, each controlling whisker movements along the rostral-caudal (R-C) axis. All whiskers received the same Gaussian white noise deflections (in position). Multi-shank silicon probes are placed in either wS2 or wS1. Bottom: The spike-triggered covariance (STC) technique involves tabulating stimulus shapes preceding each spike from a single unit (referred to as the spike-triggered ensemble, STE). A covariance based, PCA-like analysis is then applied to the STE to recover the stimulus shapes, or filters, most correlated with spiking. The same process is applied to shuffled spike trains to determine which filters are significant (See Methods). **b**, **c** PCA carried out on the populations of significant filters in wS2 and wS1. **b** Left: The top two PCs for the wS2 population explain 88.96% of the variance in the wS2 filters. Right: The top two eigenvectors for the wS2 significant filters. **c** Left: The top two PCs for the wS1 population explain 86.77% of the variance in the wS1 filters. Right: The top two eigenvectors for the wS1 significant filters. **d**, **e** Significant filters from each cortical region are projected into the two-dimensional space generated by the first two PCs of their respective regions. Filled circles: first two significant filters for each neuron; open circles: remaining significant filters for each neuron. Phase is color-coded representing alignment to the first (blue) or second (red) PC from a given region. **d** 411 significant filters from 204 neurons out of 1157 single units measured in wS2. **e** 1320 significant filters from 567 neurons out of 1038 single units recorded in wS1. **f** Phase aligning first and second PCs across regions shows an increased filter amplitude at longer temporal delays for wS2 (solid lines), and for wS1 (dashed lines) there is higher filter energy near the spike. Arrows illustrate differences. **g** Energy envelopes (black) of the relevant subspaces consider all phases in the space (wS2 solid lines, wS1 dashed lines, color coded as in **d**, **e**) **h** Left: The energy envelopes for wS2 (solid lines) and wS1 (dashed lines), the half-width ($hw_{S2}$ and $hw_{S1}$) and half-area time ($t_{S2}$ and $t_{S1}$) are illustrated. Right top: The half-area time is later in wS2 by 4.5 ms ($p < 0.001$, permutation test). Right bottom: The half-width in wS2 is wider than wS1 by ~ 6 ms ($p < 0.001$, permutation test). The rat image in this figure is adapted from Diamond et al.[1], Nature Reviews Neuroscience 9(8). All rights reserved

**wS2 has less feature-selective neurons than wS1**. The standard next step in the STC approach is to evaluate the selectivity of the neurons in the filter space. Thus, we computed the non-linear functions that map stimulus input to spiking output in the phase spaces shown in Fig. 1. The advantage of these phase spaces, or relevant spaces as we will refer to them from now on, is that we can generate these non-linear functions in a common subspace for every cell. Using this technique, it has been shown that wS1 presents simple/complex dichotomy[17]. Simple cells are tuned to a small phase region in the subspace, which reflects a precise whisker movement in time. Complex cells fire equally well to all whisker movements that have high magnitude projections into the space, a property called phase-invariance.

We found that 12.7% of the wS2 cells and 29.4% of the wS1 cells that gave significant filters were tuned to a small phase

region. This proportion of phase-tuned cells in wS1 agrees with a prior study[17] (Fig. 2a–c, see Methods). The half-width of the phase tuning among phase-selective cells differs between wS2 and wS1, with wS1 cells on average more sharply tuned as shown in Supplementary Fig. 1a (wS1 average half-width = 64° and wS2 average half-width = 80°, $p$-value = 0.018, permutation test). Thus, precise whisker movements are not as well represented in wS2 as they are in wS1. In addition to phase tuning, another cell-type that we observed was tuned to one phase and to its opposite. We call these orientation cells (Fig. 2d, e). We found that 2.5% of the wS2 cells are tuned to orientation and 6.2% of the wS1 cells ($p < 0.001$, Fig. 2f). These cells respond precisely to high velocity movements in either direction, since opposite phases correspond to an inversion of the movement direction (black filter shapes in Fig. 2d, e).

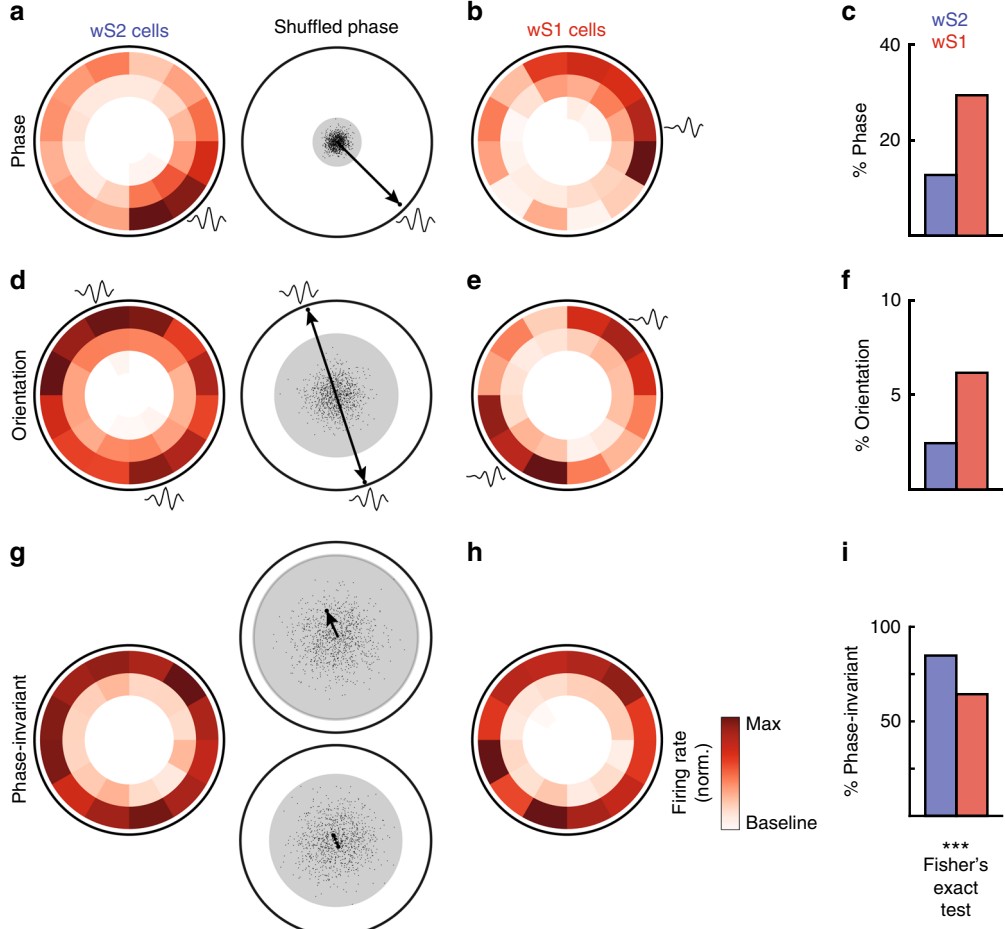

**Fig. 2** Phase selectivity is less pronounced in wS2 compared to wS1. **a** Left: An example wS2 cell with selectivity for a region in the wS2 relevant subspace. Right: A graphical representation of the procedure for evaluating statistical significance. The vector sum (arrow) of all spike-eliciting stimuli is well outside of the phase-randomized distribution (dots, the gray zone encloses all 1000 shuffles, $p < 0.001$). The black stimulus shape at the border of the region is the preferred central phase. **b** An example wS1 cell with phase tuning as in **a**. **c** A regional comparison of the percentage of cells exhibiting phase tuning (wS2 = 12.7 and wS1 = 29.5%). **d** Left: An example wS2 cell that has selectivity for an orientation in the wS2 relevant subspace, which means it is equally tuned to two opposite phases. Right: a graphical representation of the procedure for evaluating statistical significance as in **a**. **e** An example wS1 neuron with orientation tuning displayed as in **d**. **f** A regional comparison of the percentage of cells exhibiting orientation tuning (wS2 = 2.5 and wS1 = 6.2%). **g** Left: An example wS2 cell with phase-invariant tuning in the wS2 relevant subspace. Right: A graphical representation of the procedure for evaluating statistical significance. This neuron fails both phase (top) and orientation-randomized shuffling (bottom). **h** An example wS1 cell with phase-invariant tuning displayed as in **g**. **i** A regional comparison of the percentage of cells exhibiting phase-invariant tuning (wS2 = 84.8 and wS1 = 64.4%). The population differences are highly significant (Fisher's exact test, $p = 6.7e{-}8$)

Finally, we quantified the percentage of cells that were not specifically tuned to any phase or combination of phases in the subspace (Fig. 2g, h). These phase-invariant cells were much more prevalent in wS2 than wS1 (84.8% in wS2 vs. 64.4% in wS1, Fig. 2i). Comparing the percent of phase-tuned (either phase or orientation-selective) cells in the two regions shows that in wS1, twice as many cells in the population encode a precise temporal pattern of whisker movement during fully correlated stimulation (15.2% in wS2 and 35.6% in wS1). These population differences are highly significant (Fisher's exact test, $p = 6.7e{-}8$), and show that wS2 represents less low-level whisker kinematic information, or as we call it from now on, fine features.

Another important insight that can be made with these non-linear functions can provide an explanation for why we get so many more filters in wS1 than in wS2. Phase-invariant wS1 cells exhibit a much larger number of spikes per representative filter-like stimulus, as illustrated by the population average one-dimensional firing rate functions shown in Supplementary Fig. 1b.

This suggests that wS2 cells code the stimulus more sparsely than what is found in wS1.

**wS2 has a distinct dynamic stimulus dependence from wS1.** The analysis up until now was focused on the relevant phase spaces shown in Fig. 1. However, there are some filters (~10%) that do not fall near the unit circles (open circles in the subspace plots in Fig. 1b, c). This is because STC analysis requires that all significant filters that come from a single neuron are orthogonal. Neurons with more than two significant filters have some that are not well-described by the relevant subspace. The remaining filters can be described in a complementary two-dimensional subspace for each region, which resembles a time-shifted version of the relevant subspace as illustrated in Fig. 3a for wS2 (more details in Supplementary Fig. 2). This suggests that for some neurons these filters extend the temporal integration window, retaining the same shape of the relevant filter space. To capture the complete

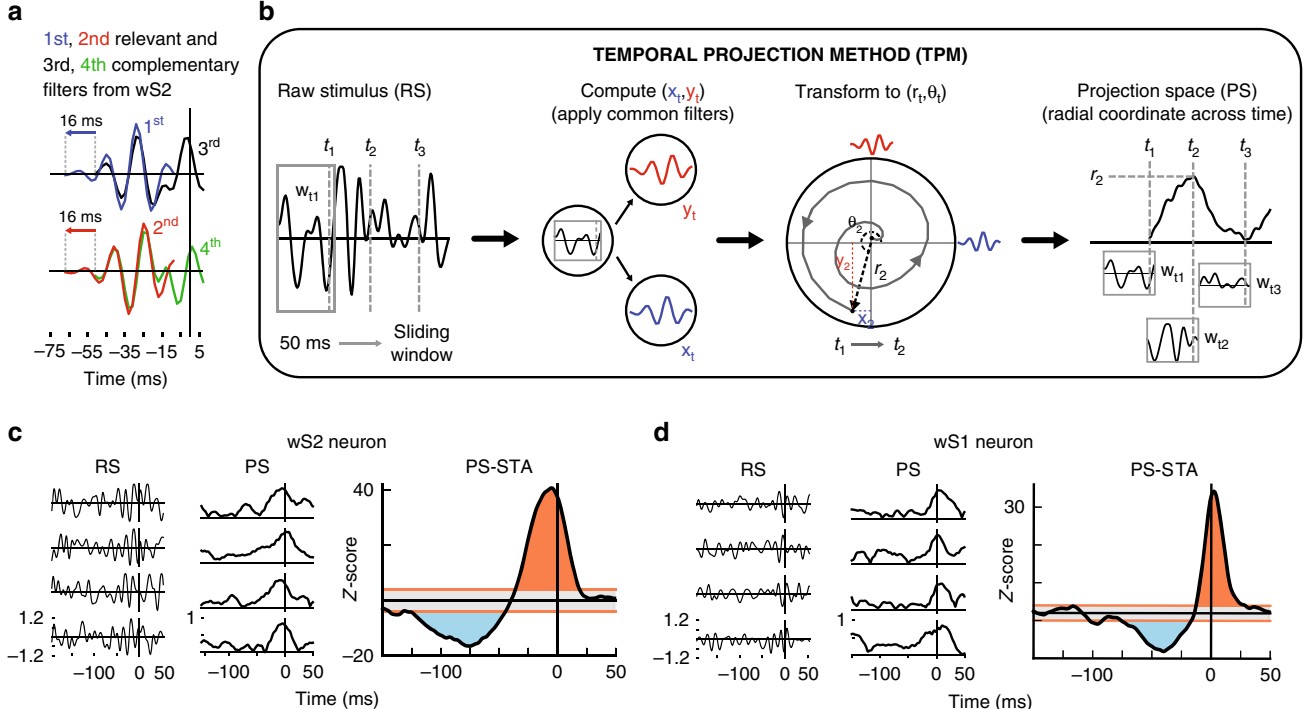

**Fig. 3** Temporal projection method (TPM) uncovers distinct temporal dynamics in wS2 and wS1. **a** Same analysis as in Fig. 1 with an extended window (−55 pre- and 5 ms post-spike) for wS2. Shifting the first and second PCs by 16 ms overlays them very well with the third and fourth PCs (See Supplementary Fig. 2). **b** Change of coordinates from whisker position raw stimulus (RS) to relevant filter projection subspace (PS). Left: A 50 ms sliding window is passed over the RS. For each time $t$, we use −45 ms pre- 5 ms post, just as the filters were generated in Fig. 1. Middle left: Each RS window can be projected into the region-specific phase space resulting in an x-coordinate on the PC1 axis (blue, $x_t$) and a y-coordinate on PC2 axis (red, $y_t$). Middle right: As the window slides through time, this creates a trajectory in phase space (($x_t$, $y_t$) or ($r_t$, $\theta_t$)). Right: The radial coordinate in the phase space across time. PS value represents how strongly the stimulus in the 50 ms window projected into the phase space. **c**, **d** Left: four examples of a spike-triggering RS. (195 ms pre- and 55 ms post-spike). Middle: PS trajectory of the same stimulus. Right: PS spike-triggered average (PS-STA) in z-score values (compared to a shuffled stimulus distribution). Gray region represents ± 2 z-score zone outside of which the area under the curve is shaded red for positive area and blue for negative area. **c** An example wS2 neuron with extended temporal stimulus dependence. **d** An example wS1 neuron that responds sharply at short latencies to high PS movements

stimulus dependence of spiking activity across both relevant and complementary subspaces, and thus obtain temporal dynamics, we looked at the evolution in time of the spike-eliciting stimuli in the relevant subspace.

To this end, we applied the change of coordinates procedure illustrated in Fig. 3b. It takes as input a 50 ms window of rostro-caudal whisker movement (the raw stimulus) and transforms it into a point in the region-specific relevant subspace. The coordinates of this point are computed as scalar products between the raw stimulus and the two relevant subspace filters. This procedure is a transformation from whisker position coordinates to a new coordinate system which is empirically determined to be most relevant for the neurons. Sliding the 50 ms window across the stimulus produces an evolution in time in the relevant subspace, which can be equivalently represented as Cartesian (x(t), y(t)) or polar (r(t), theta(t)). Polar coordinates are more convenient because the spiking activity for neurons in both wS2 and wS1 occurs predominantly at high values of the radial coordinate, which correspond to relevant filter-like whisker movements, as shown in Fig. 2. We proceeded with only the radial coordinate to further compare wS2 and wS1. Applying the temporal projection method (TPM) described in Fig. 3b to the spike-eliciting stimulus sequences, we computed the spike-triggered average (STA) of the radial coordinate in a 200 ms window around the spike times (150 ms before and 50 ms after). To test for statistical significance, we used a z-score based on shuffled data (see Methods). These projection space STAs (PS-

STAs) reveal a more elaborate picture of the temporal stimulus dependence of the evoked neuronal response than classical STC (Fig. 3c, d). The full dependence on the stimulus not only includes high radial projections just before the spike, as expected based on the STC analysis, but there is also an extended low-radial-projection tail. Low-projection movements can be low magnitude movements (window 3 in Fig. 3b), or also movements that do not project well into the space (window 1 in Fig. 3b). These tails were not present in all neurons, but a subpopulation analysis of the neurons with them showed that they have both larger strength (integral of curve) and longer duration in wS2 than in wS1 (Supplementary Fig. 3h–i). For each example cell in Fig. 3c, d, we show 4 individual spike-eliciting stimuli in both raw stimulus space and projection space to clarify the link between them (population averages in Supplementary Fig. 3a–b).

The use of TPM with continuous whisker movements uncovers extended stimulus dependencies for many neurons that yield no significant filters using classical STC. In wS2, we find 679 neurons with significant PS-STAs, while we were only able to recover STC filters for 204 of these cells. For wS1, we detect 828 PS-STAs from the same population of 1038 neurons that yielded filters for 567 cells. The cells that have significant PS-STAs but do not show significant STC filters are still well-tuned to the relevant subspaces and exhibit temporal dynamics similar to the STC filter-producing cells (Supplementary Fig. 3c–g). TPM is much less data hungry than the classical STC analysis, and it can partially compensate for the fact that wS2 responses, although

present, seem to be less strong than wS1 responses in terms of spikes/filter-like stimulus (Supplementary Fig. 1 and 3). The reason why the TPM is more sensitive at detecting functional responses than classical STC is that once the stimulus subspace of interest for the cells in a region is known, we can focus our analysis in this subspace, thus increasing the signal-to-noise ratio on which we base our functional assessment. This added sensitivity, coupled with its applicability across longer time windows than STC make it a useful tool to obtain rich dynamic stimulus dependencies.

**TPM uncovers whisker-pad scale spatio-temporal dynamics.** All analysis up until this point has focused on fully correlated multi-whisker movements, where the classical STC approach was successful. However, during a different stimulation pattern where each whisker receives a unique, uncorrelated Gaussian white noise stimulus, the STC approach yields 12 significant filters for wS2 (20 if each whisker is processed independently, see Methods). The paucity of significant filters is unique to wS2, as the single

whisker approach yielded 680 significant monovibrissal filters in wS1[17]. This suggests that many wS2 neurons compute tactile statistics at a spatial scale that is larger than a single whisker during continuous multi-whisker movements.

To investigate the whisker-pad scale spatio-temporal dynamics in wS2 and wS1, we took advantage of the TPM. Using the relevant subspaces derived from the correlated stimulation and projecting every whisker's raw stimulus trajectory into the respective relevant subspace allowed the identification of whisker-pad scale spatio-temporal receptive fields during uncorrelated stimulation. These receptive fields reveal remarkable spatio-temporal dynamics that are exemplified in Supplementary movies 1–10. These movies emphasize that important information is present beyond the static feature detection assumed by STC analysis. The stimulus dependence is not fully described within a small temporal window before a spike. In Fig. 4a, b, the top panels show example dynamic receptive fields of neurons from both wS2 and wS1, and the bottom panels display the classical receptive fields of the same cells obtained with sparse noise stimuli. There is a correspondence between the whiskers

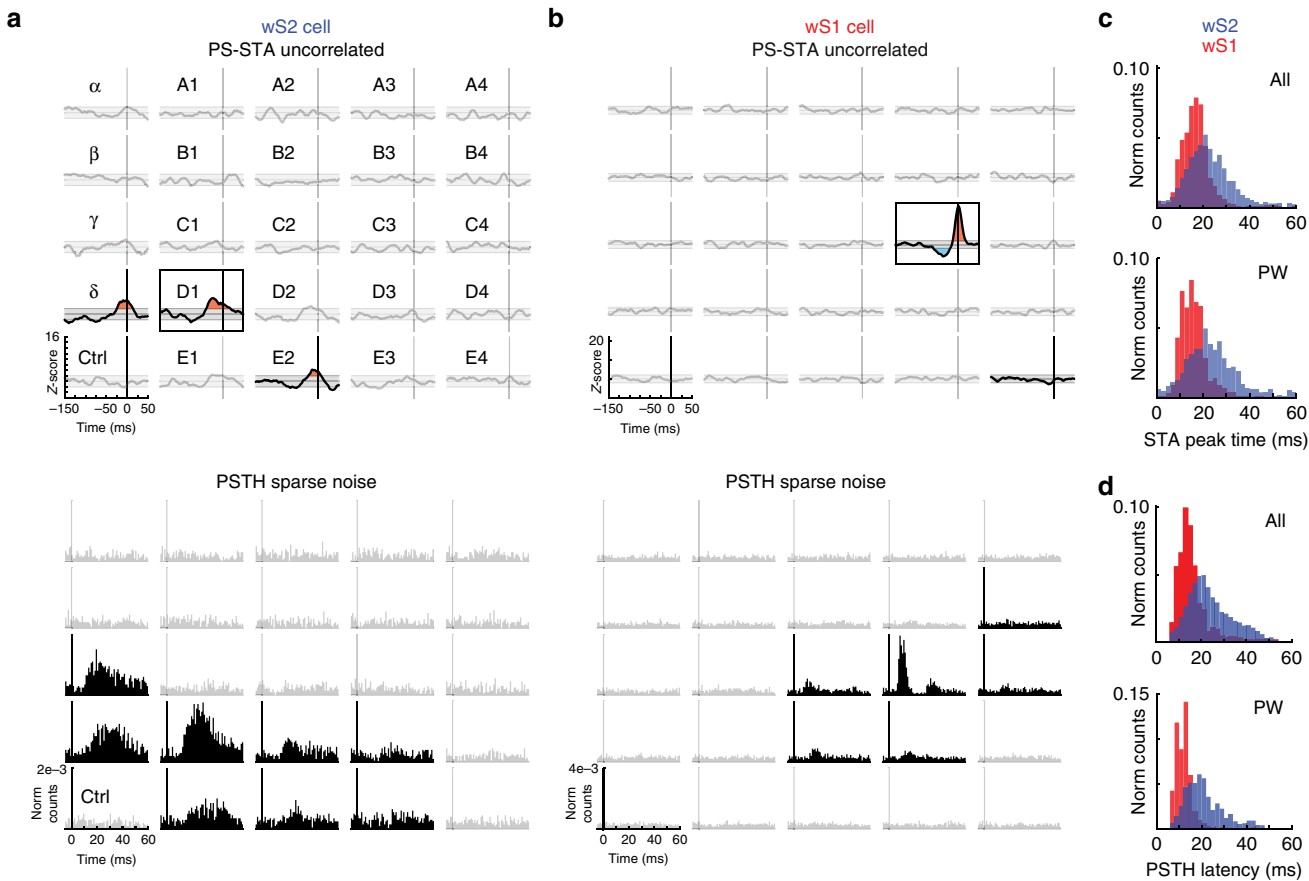

**Fig. 4** TPM reveals whisker-pad scale spatio-temporal dynamics. **a**, **b** Top: PS-STAs under continuous uncorrelated Gaussian white noise for the same neurons shown in Fig. 3. Each whisker subplot is z-scored as in Fig. 3c, d. Non-faded subplots are significant responses based on the windowed energy compared to a shuffled stimulus distribution (see Methods). Significant responses are shaded for high-PS tuning (red) and low PS tuning (blue). Black boxes delineate the strongest whisker (largest integral of the curve). A to E: whisker rows, Greek letters: straddlers, 1 to 4: whisker arcs. Bottom: Receptive fields mapped using sparse noise. Each whisker subplot is a PSTH aligned to the stimulus (including stimulus from caudal and rostral deflections for each whisker). Non-faded subplots are the ones with significant PSTH responses compared with the blank (bottom left) based on a surprise analysis (see Methods) for each direction. **a** wS2 cell. **b** wS1 cell. **c** The peak time of the PS-STAs for the populations in wS2 and wS1 shown for all whiskers (All) in the top plot and the principal whiskers (PW) in the bottom plot. The wS2 peak times are adjusted by the $\Delta t$ found in Fig. 1g for a realistic depiction of the distribution compared with wS1. **d** Latency distributions for all cells from wS2 and wS1 for all whiskers (top) and principal whiskers (PW, bottom). As expected, these distributions match well with **c**. Latencies in wS2 for all whiskers are 23 ± 10 ms (median 21 ms) and for the PWs only are 20 ± 9 ms (median 18 ms) for 681 cells and 5489 whiskers. In wS1, latencies are 14 ± 7 ms for all whiskers (median 12.5 ms) and 10 ± 4 ms for PWs only (median 10 ms) for 152 cells and 994 whiskers. All ± values are standard deviations

eliciting significant responses from the two stimulation types and between the peak times for the uncorrelated PS-STAs (Fig. 4c) and the latencies for the sparse noise (Fig. 4d). Comparing these response properties shows more temporal spread in wS2 than wS1, which agrees with the wider energy envelopes seen in wS2 (Fig. 1). As with the correlated stimulation pattern, the TPM greatly increased the number of detected functional responses during uncorrelated stimulation. We detected 735 responsive neurons from wS2 (vs 5 with STC) and 870 from wS1 (vs 429 neurons with STC).

**TPM reveals distinct specialization in wS2 and wS1.** The information present in the dynamic receptive fields shown in Fig. 4a, b allows a detailed comparison of the full spatio-temporal stimulus dependencies in wS2 and wS1. To establish this comparison, we quantified stimulus-response properties, detailed in Supplementary Fig. 4, that were chosen to be reliable features of the PS-STA curves. These include peak times, spread (width and extrema time difference), and peak sharpness for both the correlated and uncorrelated stimulation patterns, as well as their ratios between the two types of Gaussian white noise stimulation, and the number of whiskers impacting neuronal firing in the uncorrelated stimulation.

Some of the quantified features show differences in their distributions between wS2 and wS1. For example, the quarter-width of the PS-STA computed for correlated stimulations confirms the extended temporal integration windows for wS2 that were described in Fig. 1g, h (Fig. 5a, top). The sharpness of the response to the whisker eliciting the strongest PS-STA during uncorrelated stimulations reflects how precisely locked a response is to a relevant whisker movement (Fig. 5a, bottom). Sharp PS-STAs likely correspond to fine feature selectivity, and these properties are significantly less pronounced in wS2 than in wS1 (Fig. 2 and Fig. 5a).

Given that there are differences in the distributions of individual response characteristics between wS2 and wS1, we quantified how they separate across all the measured response

properties. We summarized the differences using PCA, which helps to visualize the degree of specialization between wS2 and wS1 (Fig. 5b, c, the single feature basis and PCA weights are provided in Supplementary Fig. 4). The PCA shows clear specialized regions for wS2 and wS1, reflecting the unique parts of the representations in each cortical area. Different cell densities in different regions of the coding space imply that higher percentages of the respective populations occupy certain functional regimes. The separation between wS2 and wS1 lies mainly along the first principal component which is derived from the spread, latency, and sharpness of the evoked responses in both correlated and uncorrelated stimulation patterns (magenta bars in Supplementary Fig. 4n). The second principal component is related to the ratios between responses to the correlated and uncorrelated stimulation (cyan bars in Supplementary Fig 4n). The bigger spread in the second principal component of wS2 compared to wS1 emphasizes the diversity of the different responses across the two stimulation patterns.

**Multi-whisker integration is supra-linear in wS2.** Beyond temporal features, the PS-STAs also provide information about spatial integration. To extract this, we compared the sum of the response strengths (see Methods) of single whiskers obtained during the uncorrelated stimulation to the response strength obtained during correlated stimulation (Fig. 6). This ratio indicates whether inter-whisker correlation in the stimulus increases the response of a neuron more than what would be expected by linear summation across whiskers.

An exemplary wS2 cell (Fig. 6a, left) has a broad temporal response profile for the correlated stimulation and a much smaller response to its principal whisker during uncorrelated stimulation (but also broad). This is a neuron that supra-linearly integrates correlated whisker movements across space. In contrast, the characteristic wS1 cell shown in Fig. 6b (left) has a sharp, fast temporal response for both correlated and uncorrelated stimulations, and a slightly supra-linear integration across whiskers. The principal whisker evoked response during

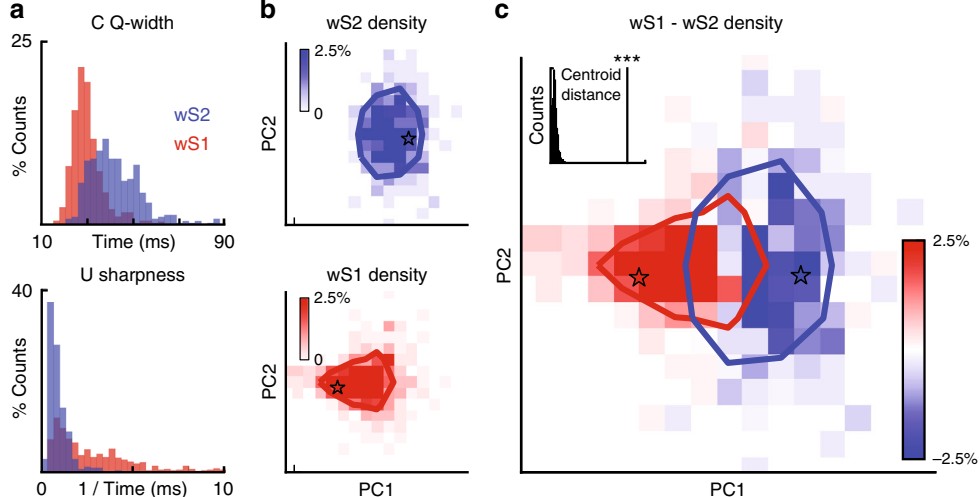

**Fig. 5** TPM reveals wS2 and wS1 specialization. **a** Top: correlated quarter-width feature (Q-width) of PS-STA across all responsive neurons in wS2 and wS1. Bottom: the sharpness (height/width) of the PS-STA to the whisker eliciting the strongest response during uncorrelated stimulation. All quantified PS-STA features are detailed in Supplementary Fig. 4. **b** Top: wS2 density plot across the first two principal components of all quantified PS-STA features. Bottom: wS1 density plot across the first two principal components of all quantified PS-STA features. Solid lines are contours (blue wS2 and red wS1) with everything inside at greater than ¼ of the maximum density. **c** wS1–wS2 density plot (blue for higher wS2 densities and red for higher wS1 densities) in feature space shows specialized zones where there is predominance of neurons from each cortical area. Stars (blue for wS2 cell and red for wS1 cell) show location of the example cells from Figs. 3 and 4

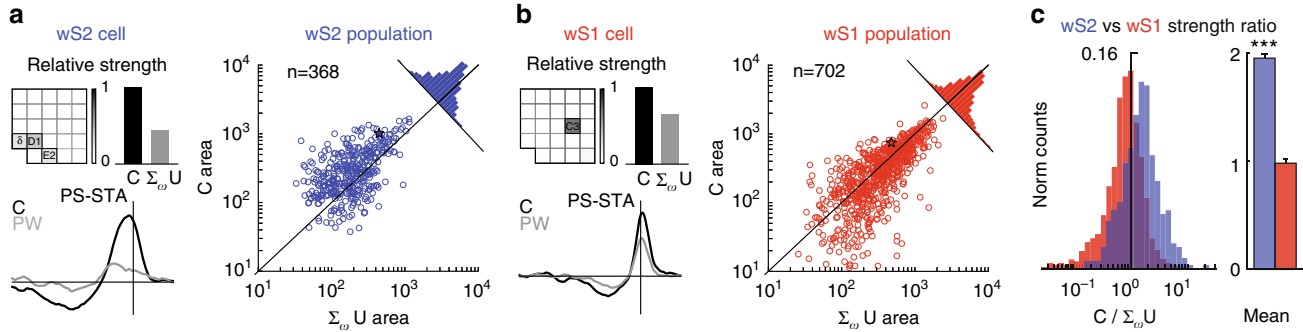

**Fig. 6** Supra-linear multi-whisker integration in wS2. **a** Left: an example wS2 neuron (the same cell from Figs. 3–5), Left Top: whisker pad heatmap (same arrangement as in Fig. 4a) and bar plot showing the relative response strength (integral of PS-STA) of the neuron to individual whiskers during uncorrelated stimulation compared to the response strength to correlated stimulation, Bottom: The PS-STA for the principal whisker (light gray curve) during uncorrelated stimulation and the correlated stimulation (black curve). Right: the population distribution of neurons in wS2 with the sum of the individual whisker strengths from the uncorrelated stimulation (log scale) on the *x*-axis and the strength of the correlated response (log scale) on the *y*-axis. The star on the plot corresponds to the example cell and the histogram on the top right illustrates the distribution of the ratios. **b** Same as **a** but for a wS1 example neuron (same neuron as shown in Figs. 3–5) and the wS1 population. **c** Left: The two histograms from parts **a** and **b** overlaid. Right: A bar plot comparing the mean strength ratios between the correlated stimulation and the sum of individual whiskers in the uncorrelated stimulation. Significance ($p < 0.001$) assessed by shuffling cell/region mappings and computing difference in means. Bar plot whiskers are standard errors

uncorrelated stimulation is slightly smaller than the response in the correlated stimulation.

The multi-whisker response of wS2 neurons is more than a simple summation of the single whisker influences, as shown by the many neurons in wS2 that fall above the diagonal in Fig. 6a (right). This supra-linearity is much more pronounced in wS2 than in wS1 (Fig. 6b, right). The wS1 population falls very close to the diagonal of the plot which suggests mainly linear integration across whiskers, as is reflected by the histograms and mean comparisons depicted in Fig. 6c. The response strength differences are not caused by a difference in firing rate in the regions between the different stimulation patterns (Supplementary Fig. 5). These results demonstrate that inter-whisker correlations generally reinforce the strength of the response in wS2 more than in wS1. This suggests that wS2 carries out a computation relating to the multi-whisker statistics of the tactile scene.

## Discussion

Our work demonstrates that wS2 and wS1 cortical areas in the rat contain specialized representations that likely support complementary and interdependent aspects of tactile sensation. In the visual system, it is generally accepted that as early as the retina, different types of visual information are extracted and sent through many processing channels, and this is reflected by a visual cortex parcellated into specialized areas[18]. By applying three stimulation patterns with different statistical properties and carefully comparing evoked responses in wS2 and wS1, we conclude that the somatosensory system follows the same general principles. wS1 encodes precise temporal stimulus features occurring at a single whisker scale (Figs. 2 and 4). The responses in wS2 depend to a much larger degree on multi-whisker correlation, they integrate whisker movements over longer temporal windows (Figs. 1 and 5) and exhibit supra-linear responses to correlated multi-whisker movements (Fig. 6). If the single whisker contributions to the firing of a single neuron are temporally very sharp as in wS1, the coding space would explode in terms of multi-whisker spatio-temporal combinations. The wS2 neurons sum inputs over longer temporal windows to more efficiently encode multi-whisker synergies. Broad-scale textural statistics, object-shape configurations, and global motion are examples of the computations that can be made with multiple spatial inputs sampled across extended windows of time. The functional

properties that we found in wS2 cells are consistent with the ability to perform these global types of computations.

What does this mean regarding the hierarchical organization of wS2 and wS1? In the rat whisker system, the mechanoreceptor connections with the neurons in the principalis nucleus (PrV) of the brainstem preserve single whisker, temporally precise receptive field properties in this area[19–21], which then are passed along the lemniscal pathway all the way to wS1. In a parallel manner, multi-whisker information is gathered through the broadly branching dendrites in the interpolaris region of the spinal nucleus (SpVi) of the brainstem and routed through the paralemniscal and extralemniscal pathways to the cortex[22,23]. wS2 seems to be the preferred destination for extralemniscal projections, although there are weaker projections to the septa in wS1, and paralemniscal projections equally target wS2 and wS1[24].

Our results confirm that the functional segregation that originates in the brainstem is also reflected in the coding properties found in wS2 and wS1. The stimulus feature encoding along these pathways has been studied using both forward and reverse correlation techniques. Reverse correlation has revealed increasingly complex and diverse feature representations arising between the mechanoreceptors and wS1[17,25,26]. Our results show that very few of the cells recorded from wS2 can be easily characterized as responding to a single stimulus feature. Phase and orientation tuning in our relevant subspace model of wS2 were much rarer than for wS1, and tuned wS2 cells are relatively less tuned than tuned wS1 cells. While taking a classical feed-forward view would clearly point to wS2 as a higher cortical area based on its increased phase invariance, if this is taken together with the anatomical connectivity, it looks more likely to be the hub of cortical processing for a different kind of whisker information than what is found in wS1.

From a perceptual standpoint, intact somatosensory cortical processing in both wS2 and wS1 is required even for single whisker detection tasks in mice[14,27,28]. Thus, it might be very difficult to find a perceptual task that can tease apart the respective sensory contributions of the two areas. In our anesthetized preparation, we can precisely control the stimulus applied to every whisker, which makes it possible to vary global (inter-whisker) and local (single whisker) statistical properties of the stimulus independently. This would not be possible in awake, behaving animals as they would actively palpate objects and the

statistics of the input at the level of the whisker follicles would be behavior-dependent.

The anatomical and functional details of the whisker cortical areas in rats that we have described elucidate the complementary and interdependent nature of the whisker information streams. Conceptually, fine feature encoding can be enhanced by accounting for the global statistics of the tactile scene and the global tactile scene can depend on the collection of fine features encountered in the recent history. Until now, these specialized functional representations were only theorized[6,15,29,30], and not assigned to specific cortical regions. We present the first evidence that maps these functions to the two largest somatosensory cortical regions in the rat. Within these regions, there is also redundancy in the representations, which is not surprising considering they are highly interconnected and share thalamic inputs. The specialized representations found in the two whisker cortical regions provide important insights into how rodents parse the tactile world.

## Methods

**Animal procedures**. Experiments were performed in conformity with the French Ethical Committee (Direction générale de la recherche et de l'innovation) and European legislation (2010/63/EU). Animals were male Wistar rats ($N = 54$, 310 ± 42 g). Anesthetic induction was made with 3% isoflurane mixture in 80% $N_2O$ and 20% $O_2$ delivered at 1 L/min before the animals were placed into a stereotactic device. Body temperature was maintained at 37° with a feedback-controlled heating pad connected to a rectal probe. Eyes were protected by applying an optical gel (Opthalon). Anesthetic state was assessed throughout the whole experiment by means of an ECoG electrode placed on the surface of the brain through a small craniotomy (~0.5 mm) placed anterior to the multi-electrode recording site. A craniotomy of about 2 × 2 mm on either the left posterio-medial wS1, or wS2 was made in the same hemisphere to expose the brain surface (6.0 mm lateral–3.7 mm posterior to Bregma for wS1 and 7.5 mm lateral–2.8 mm posterior for wS2). wS1 experiments were already reported in a previous work[17] and here we perform new analyses on the data obtained. To access wS2, the temporalis muscle was stretched laterally, and the probe descent was made with an angle of 65° with respect to the surface of the brain. After removing the dura, multi-site silicon probes (Neuronexus 64 channels Buszaki64 electrodes, 160 $\mu m^2$ electrode size in 4 or 8 shanks geometries for wS2, or 32 channels linear, 177 $\mu m^2$ electrode size for wS1) were lowered to reach depths between 700 and 1800 $\mu m$ (1000 and 1600 $\mu m$ for wS1). The reason to aim at granular and infragranular layers is the need of sufficient spiking activity to carry out STC analysis (see Data analysis below). At this moment, anesthetic state was changed to light (stage III, plane 1–2) by gradually decreasing the isoflurane concentration. The level of anesthesia was assessed by 1) respiration rate between 1–1.5 Hz, 2) lack of spontaneous movements and 3) presence of fast ECoG oscillations (> 5 Hz).

**Histology**. In all wS2 experiments and half of the wS1 experiments, DiI was deposited on the shanks of the electrode[31]. At the end of the experiment, a pentobarbital overdose was injected (ip) and the animal was perfused transcardially with saline and then 4% formaldehyde solution. Either coronal slices of 90 $\mu m$ thickness or flattened cortical slices of 46 $\mu m$ thickness were cut and stained with cytochrome oxidase to visualize simultaneously electrode placement and layer IV barrels in wS1 with different perspectives (Supplementary Fig. 6). Since there are no barrel structures in wS2, we confirmed that the electrode recording sites were away from the posterio-medial barrel cortex.

**Electrophysiological recordings and clustering method**. Raw electrophysiological traces were acquired at 30 kHz. They were processed offline, using the Klusta suite[32]. Signal was filtered between 500 Hz and 3 kHz with a third order Butterworth filter. 48 samples from each waveform were saved to do the spike sorting. Automatic sorting of waveforms was run by the Klusta suite, and a second step of data curation was made manually. As a first step in the manual cleaning, signal to noise ratio (SNR) was evaluated for each triggered spike with respect to root mean square noise of the recorded traces. Spikes with SNRs < 3 on all recording sites were discarded. To aid with the quantification of our manual curation, we computed the $L_{ratio}$ and the Isolation distance ($I_d$)[33], which offer an advanced measure for multi-channel recordings to evaluate cluster quality. $L_{ratio}$ measures the compactness of a cluster, with low values meaning low contribution from non-cluster spikes and highlights type II (false negative) error types, whereas $I_d$ measures the distance of the $n^{th}$ closest noise spike if a cluster has n spikes, and highlights type I (false positive) error rates. The metrics for our population of wS2 cells allowed us to focus on well isolated units for our analysis.

**Stimulus application**. The 24 caudal vibrissae from the right snout of the rat were cut to 10 mm length and then placed inside the tip of each piezo-electric deflector of a custom-built whisker stimulator[16] (Supplementary Fig. 7a left). The position of each piezo-electric bender matched the resting position of each whisker, which was stimulated 7 mm from the base. The resting position was defined as the angle 0° for the rostro-caudal direction (Supplementary Fig. 7a, right). The stimulation protocol included three different patterns, two with continuous Gaussian white noise stimulation and one with sparse noise stimulation, with each stimulus repetition lasting 10 s. The order of stimulation of the three patterns of stimuli was randomized during 2.5 h of measurements for each recording (Supplementary Fig. 7b). Gaussian white noise stimuli were built by selecting from a Gaussian distribution of positions with mean 0 and standard deviation of 5000 at 5.5 ms intervals. These points were then connected by cubic splines and smoothed to obtain a whisker deflection command rate of 1 kHz that fell within the technical specifications of our piezo-electric benders. Distribution tails were cut to avoid possible ringing from large deflections[16]. For the correlated deflections all whiskers had the exact same synchronous movement and for the uncorrelated every whisker had a unique Gaussian white noise. Sparse noise stimulations consisted of randomly stimulating one whisker at a time in either the rostral or the caudal direction with a stimulus profile that was a 10–10–10 ms ramp-hold-ramp with an amplitude identical to the maximum in the Gaussian white noise (Supplementary Fig. 7b).

The stimulation command was followed faithfully by the stimulator as can be seen in Supplementary Fig. 7c ($r = 0.997$) when measured with laser telemetry (MicroEpsilon). The angular position and velocity of the whiskers sampled across the stimulus can be seen in Supplementary Fig. 7d. (−1.2° to 1.2° angle range and 440°/s maximum speed), which match the range of values observed during contacts with textures in freely behaving rats[34]. The frequency content of our experimental stimulus followed closely the input command and had a flat 3 dB band stretching up to 79 Hz (Supplementary Fig. 7e). To make sure that the stimuli applied during wS2 and wS1 recordings did not have any differences in temporal correlation, we computed the autocorrelation of the stimulus used in both wS2 and wS1 experiments. We found a full width half maximum of 7 ms for both (Supplementary Fig. 7e).

**Transient onset responses**. At the onset of each sweep of Gaussian white noise stimulus, there was a global increase in firing that quickly attenuated. Before making STC analyses, all spikes in the time window including this transient response were discarded. This enhances the power of the STC method because the system is steadily in the adapted state and is firing to stimulus features rather than to a large contextual change. To compute the duration of this transient response, we constructed a peri-stimulus-time-histogram (PSTH) of the spiking activity of each neuron with respect to the onset of each stimulation pattern. We found the mean firing rate $\mu$ and its standard deviation $\sigma$ at a stationary state (between 0,75 and 1 s after stimulus onset) and the peak of the PSTH with 10 ms bins. We then performed a forward search in the PSTH until three consecutive bins gave values below $\mu + 3 \sigma$. Population data in the wS2 correlated stimulation rendered an offset value of 150 ms after stimulus onset leaving only 4 neurons out of 1157 with responses extending slightly out of this window. The uncorrelated stimulus offset times followed closely the ones in the correlated stimulus.

**Spike triggered covariance method (STC)**. All off-line data analyses were performed with programs coded in the Python language. Reverse correlation analyses were made on spikes recorded during Gaussian white noise stimulation. In brief, STC analysis aims to find the relevant subspace of the stimulus ensemble, or bank of linear filters, that best captures the variations in the stimulus leading to the spiking of a neuron. The filter shapes are useful for understanding the features of the stimulus encoded in a sensory region[35,36]. These reverse correlation methods have not yet been applied in wS2. To obtain the filters, an inspection window containing pre-spike stimulus is determined first, which in our case was chosen to be 50 ms (45 ms pre- and 5 ms after-spike). Because our stimulus contained temporal correlation, we were able to down-sample in 2 ms steps without losing information, obtaining windows with 25 dimensions (points in time). For the correlated stimulation, the only dimensions that vary are in temporal dimensions. There is only one spatial dimension because every whisker receives the same stimulus. After extracting the spike-triggered ensemble of stimulus shapes, the filters are obtained by a principal component analysis (PCA) of the whitened covariance matrix and using a regularization to avoid artificially introducing high frequencies. For this constraint we used a ridge regression homogeneous parameter $\lambda = 5.0 \times 10^7$ for wS2 and $\lambda = 5.5 \times 10^5$ for wS1. Different values were used because the Gaussian white noise stimulus for the two regions had different variances. Changes in stimulus variance have been shown to have no effect on the filter shapes in wS1[25]. Statistical significance of a filter was determined by applying the same procedure with 200 shuffles of the spike times, retaining the inter-spike intervals of the spike train, and considered only eigenvectors from the PCA whose eigenvalues were above 8 standard deviations from the shuffled mean. Finally, the relevant filter subspace and the complementary filter subspace (see Supplementary Fig. 2) were obtained by doing a PCA on all significant filters of either wS2 or wS1. The same methods were applied in the uncorrelated stimulation but with the addition of spatial dimensions for the whiskers.

While it was possible to uncover many filters with uncorrelated Gaussian white noise in wS1 (680 filters, Estebanez et al.[17]), wS2 yielded very few filters. If all dimensions of the stimulus are considered (24 whiskers × 25 time bins), we found 12 significant filters from 5 neurons. If each whisker is processed independently (25 time bins), there were 20 significant filters from 13 neurons. These few filters fall into the same subspace as those in the correlated stimulation. The main reason for this is likely due to the requirement of an extensive amount of data for neurons with broad multi-whisker receptive fields. If we consider a minimum of 50 spikes per dimension, this gives a cutoff of 1250 spikes during correlated stimulation to carry out the STC analysis. This number gets multiplied by the number of whiskers contributing to the spiking in the uncorrelated stimulation, which is larger in wS2 than in wS1.

**Phase tuning**. The phase-tuned type of cell (Fig. 2a, b), as already described in wS1[17], was detected by randomly shuffling the phase of each spike-eliciting stimulus in the spike-triggered ensemble. If the vector sum of all spikes was outside of the null distribution (1000 shuffles, $p < 0.001$), a cell was called phase-tuned (Fig. 2b, right). The procedure to detect orientation-tuned cells is equivalent, with the addition of a previous step where we multiplied each phase by two. After calling a cell significant, the resulting vector phase is obtained by dividing by two. The 1-D nonlinear functions were obtained by averaging across the population of phase-invariant cells all the rays that cross the origin. For example, the 0°–30° bins and the 180°–210° bins were taken as the positive and negative sides of a non-linear function, and then 30°–60° bins and the 210°–240° were grouped, and so on. The resulting five functions were averaged to give a 1-D average function, which was then normalized to the basal firing rate of each cell. The phase tuning curves were obtained by centering each cell on its preferred phase, carrying out a linear interpolation to smooth and then averaging across the population of tuned cells.

**Temporal projection method (TPM)**. The TPM method goes beyond the traditional STC analysis because it can describe the behavior of neurons during continuous stimulation that do not produce STC filters. This is because it is focused on the empirically determined relevant dimensions (determined by a population analysis of filters), and thus avoids the problems associated with high dimensionality that are encountered with traditional STC. A neuron that does not have significant STC filters may still respond to stimuli with high projections into the empirically determined feature space. The 'snapshot' that is considered for STC analysis does not capture the complete picture of the stimulus dependence. There are richer temporal dynamics that need to be considered. The steps to perform this method are: first project the raw stimulus (RS) into the relevant subspace. This is done by computing the dot product of a 50 ms stimulus window with each relevant filter, which gives two coordinates. Then, this procedure can be made dynamic by sliding the 50 ms window along the stimulus and computing the coordinates at every time point to obtain a temporal evolution of the stimulus in the relevant subspace. Second, the coordinates can be expressed in polar form in the relevant subspace (projected stimulus or PS). Third, use the radial coordinate to build spike-triggered averages (STA of PS, i.e., PS-STA). Fourth, compute the null distribution from a shuffled stimulus on the same spike train. This is done by shuffling the stimulus 200 times to calculate the z-score value of the STA.

The TPM is mathematically very similar to standard linear-nonlinear Poisson models, but with important extensions. The stimulus first passes through a filter bank of two filters (products of this linear operation are $x$ and $y$), then computing the radial coordinate can be thought of as the nonlinear part ($sqrt(x^2 + y^2)$) of LNP. This can be interpreted either as a 'subspace energy' like from the Adelson energy model, or as just a vector strength. The output of this process (the radius) is then averaged across times leading up to and following all spikes ($-150$ ms to $+50$ ms). This visualization helps to see that the average radius just before spiking is higher than chance (as would be expected for neurons giving filters because this is the standard LNP), but also that is not the complete story of what causes the neuron to fire. Extending this windowing backward shows a more elaborate stimulus dependence, which varies between wS2 and wS1. Biologically, this can be interpreted to mean that when we shake the whiskers vigorously or the animal shakes them as such, there is not just a simple fixed feature processing going on, the dynamics are extended and depend on stimulus history at longer latencies.

To graphically show the response of the neurons (Fig. 3c, d) in the relevant subspace, we shaded in gray a region between $\pm 2\sigma$ from the mean of the shuffled distribution to represent the baseline, we also shaded in red the area beneath positive z-scores above the baseline and shaded in blue negative z-scores below the baseline. The method can be applied easily to either the correlated or to the uncorrelated Gaussian white noise since it only implies averaging and computing $\mu$ and $\sigma$ of a shuffled distribution of the projected stimulus (PS). Projected stimulus STAs were computed for 150 ms pre- and 50 ms post-spike to capture the full temporal dynamics during continuous stimulation.

At any moment in time our sliding window is not centered temporally: the projection is computed with 45 ms of stimulus that happened before and 5 ms that happened after, so the projections at time 0 correspond to the static relevant subspaces found in Fig. 1. Peak alignments of PS-STA to the right of the spike time do not mean prediction. They are caused by the fact that individual unit filters may be slightly narrower than the population filters used to make the subspace projections.

We computed the significance of each PS-STA comparing it to the blank stimulus (bottom left panel of uncorrelated PS-STAs, Fig. 4). For each neuron, we shuffled 1000 times the blank stimulus retaining the inter-spike intervals of the spike train and computed the z-scored PS-STA. We then used a sliding window of 25 ms across the 200 ms shape and computed the maximum positive and negative area. The small window is used to capture focused responses and not the sum of random deviations of the signal. We then regarded a correlated PS-STA as significant if after doing the same process its maximum value was bigger than 975 values of the shuffled blanks ($p < 0.025$). For the uncorrelated stimulation, we did the same for each whisker and corrected for multiple testing (24 whiskers/tests, Benjamini-Hochberg correction). We found that 943 and 942 neurons were responding either to the correlated or the uncorrelated stimulation with our method in wS2 and wS1 respectively. This supports its general applicability and capacity to capture responses in the whisker system under complex stimulation conditions, especially in the wS2 region which more sparsely encodes longer and spatially spread stimulus features.

We computed the positive (negative) strength of the PS-STA as the value of the area below (above) the curve in z-score values to 0, minus the mean of the same value from the 1000 shuffled blanks. The total strength is defined as the sum of both the positive and negative strength.

When we make a normal PSTH during sparse noise stimulation, we align all the spikes with respect to the onset of a fixed stimulus, a whisker ramp for example, and we analyze the features about where these spikes fall with respect to the fixed stimulus (latency, peak, sharpness, etc). A PS-STA inverts this idea. We align a depiction of the stimulus (how 'filter-like' is the stimulus that occurred before a spike) with respect to the spike times. When we average across all spikes, we can get an idea, compared to the average stimulus, how different is the average spike-eliciting stimulus in terms of 'filter-like-ness'. This quantification can be extended backwards in time well before the spikes occurred and gives an interesting picture of the stimulus dynamics that occur before spikes.

Characteristic features of projection space STAs both in the correlated stimulation and uncorrelated stimulation were used to show the different specializations of wS2 and wS1. For those purposes, 12 features were obtained for each responsive neuron, as can be seen in Supplementary Fig. 4. Then, a combined PCA analyses of all wS2 and wS1 neurons was performed.

**Surprise analysis**. Classical sparse noise stimulus responses were analyzed by construction of individual peri-stimulus-time-histograms (PSTH) for each whisker deflection event with a bin size of 1 ms. Two additional histograms were made for no stimulus (blank) events, giving a total of 50 PSTHs for each neuron. Time 0 was aligned to the onset of each stimulus and spike counts were obtained from all sparse noise repetitions in the experiment. To account for different firing rates across neurons, a surprise analysis method[7,37] was applied to evaluate significant responses and the response latencies. The advantage of this method is that it considers multiple timescales of response and is not sensitive to the baseline firing rate of the neuron, giving a more reliable outcome for the low firing rate neurons of wS2 and wS1. A high value of surprise corresponds to an unlikely spiking level of the neuron that is a probable functional response. Surprise is defined by: $S(f) = -\log_{10}(cdf(f, f_b))$ where $f$ is the firing rate of a PSTH bin, $cdf$ is the cumulative distribution function and $f_b$ is the baseline firing rate. The $cdf$ is obtained by using the baseline firing rate of all neurons, distinguishing between regular spiking and fast spiking cells to account for differences in firing rates, and computed for 20 different bin sizes to account for different temporal response profiles. A neuron is considered responsive to a whisker deflection direction if: (1) it has at least two consecutive bins above surprise threshold for at least one bin size, (2) it has a minimum number of counts above blank. The surprise threshold was computed as the largest between the 0.001% value in the population blank $cdf$ and the maximum value in each neuron blank surprise. A sliding window was applied to each PSTH and the maximum counts were computed for each whisker and then the maximum counts from the blank PSTH were subtracted. We considered the neuron as responsive if in wS2 the counts were at least 3 for 4 or 5 bin windows and for wS1 if the counts were at least 5 for 2 bins or 4 for 1 bin window.

**Latency detection**. Once the surprise analysis was done for the 20 bin sizes of significantly responding whiskers, we computed the latency for each of them and reported the lowest value, since different firing rates or response profiles could be detected using an optimal bin size. Surprise profiles are very sensitive and rise rapidly in the presence of a functional response. Therefore, we made a backward search in surprise profiles from its maximum value, until finding a value below the threshold in a 5 ms range. To be more precise, we made a second backward search from this value to a bin below half the threshold value.

## Data availability

The data that support the findings of this study are available from the corresponding author upon reasonable request.

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

## Acknowledgements

We would like to thank Valérie Ego-Stengel for reading the manuscript and providing comments, Aurélie Daret and Guillaume Hucher for the technical help with animals and histology. This work was supported by Equipe Fondation de la Recherche Médicale (FRM) DEQ20170336761. Matías Goldin was supported by the Paris-Saclay University (Lidex NeuroSaclay) and the European Union's Horizon 2020 research and innovation programme under the Marie Sklodowska-Curie grant agreement No 702726. Evan Harrell was supported by the Paris-Saclay University (Lidex NeuroSaclay) and by the International Human Frontier Science Program Organization (CDA-0064-2015, awarded to Brice Bathellier).

## Author contributions

M.A.G., E.R.H., and D.E.S. conceived research; M.A.G and E.R.H. performed wS2 experiments; L.E. performed wS1 experiments; M.A.G. and E.R.H analyzed the data and developed the new method; M.A.G., E.R.H., and D.E.S. wrote the paper.

## Additional information

**Competing interests:** The authors declare no competing interests.

