## [Peer Review File · Nature Communications]

Reviewers' comments:

Reviewer #1 (Remarks to the Author):

The authors address an important problem in sensory neuroscience: what are the spatial-temporal, multiwhisker receptive fields of whisker secondary somatosensory (wS2) neurons and how do they compare with those of whisker primary somatosensory (wS1) cortex neurons? Touch sensation relies on activity across somatosensory cortical areas, but in rodent the vast majority of work has focused on wS1, with wS2 remaining relatively neglected. The authors therefore apply their many-whisker stimulation device (previously described) to probe the responses of wS2 single units in anesthetized rats.

-The authors record from an impressively sample size of single units in wS2 (>1000). Although the Methods section describes wS1 recordings, the Results make it sound like wS1 results were from reanalysis of data recorded for an earlier study (please clarify!) They compare wS2 with wS1 results.

Using spike-triggered covariance-based analyses, the authors find evidence that wS2 neurons integrate over ~5 ms longer windows than wS1 neurons, and that fewer wS2 neurons show sharp tuning to features of whisker kinematics.

The authors use a custom "temporal projection method (TPM)", together with uncorrelated white noise stimuli applied to the different whiskers, to reveal additional features of wS2 responses and to compare them to wS1 responses.

Comparing stimuli in which the different whiskers moved in a correlated manner with stimuli in which they moved in an uncorrelated manner, the authors find evidence that integration of multi-whisker input in wS2 is supra-linear, more so than in wS1.

Points to address:

-I found Figure 1 hard to understand. This may be improved by adding more relevant detail to the legend.

-Regarding the authors' temporal projection method (TPM, depicted in Figure 3b): I get that, as they state, the "...procedure is a transformation from whisker position coordinates to a new coordinate system which is empirically determined to be most relevant for the neurons." But I wasn't clear about its underlying logic.

-In general, I think the paper is going to be very hard to understand for readers not already familiar with spike-triggered covariance analysis. Some effort to make it more accessible would be appreciated by many readers.

I have concerns about the Introduction, as follows:

-"Many anatomical tracing studies agree that based on the thalamic input structure and the nature of their reciprocal connectivity, wS2 and wS1 can be thought of as equally placed in the somatosensory hierarchy^{8,9}."

This is hardly a consensus view. There is a long history across multiple species of addressing this question. My own view is that S2 is more integrative and based on multiple lines of evidence situated at a higher level of the cortical hierarchy. To take one feature, wS2 sends a dense projection to layer 1 of wS1, whereas wS1-to-wS2 projections do not specifically target layer 1. For some recent

treatment of this issue in limb cortex, see:

Suter, B.A., and Shepherd, G.M. (2015). Reciprocal interareal connections to corticospinal neurons in mouse M1 and S2. *J Neurosci* 35, 2959-2974.

-“The reciprocal connections from wS2 to wS1 are symmetric.” What do the authors mean by symmetric? If they mean “homotypic” (somatotopically matched areas connecting to one another), they are definitely lacking appropriate citations (also, the Liao and Chen study cited deals with paw and not whisker areas).

-“Before making any assumptions about the roles of wS2 and wS1 in perception, it is important to adequately assess wS2’s sensory representation.” I agree that assessing wS2’s sensory representation is an important goal. But it is unfair to imply that the studies cited [11,12] made “assumptions” about the roles of wS2 and wS1---rather, the roles of these areas were experimentally investigated, not assumed. Moreover, it is far from clear that wS2’s sensory representation can be “adequately” assessed under anesthesia, and with only rostral-caudal whisker deflections, as in the current study.

I suggest that the tone of the entire introduction up to this sentence could be softened---the claims are overly strong and not adequately cited.

Reviewer #2 (Remarks to the Author):

This manuscript characterizes the feature selectivity of S2 neurons under passive whisker stimulation, with an approach similar to that previously applied by the authors in S1 barrel cortex, which led to important earlier results (Estebanez et al, *Nature Neurosci* 2012; *Nat Commun* 2016). The present manuscript extends the approach to attempt a systematic and comparative understanding of response tuning properties in S2 vs S1. Such a systematic exploration under controlled stimulation is important and needed, and complementary to the need to understand how responses are affected by active vs passive whisker exploration, and by behavioral and brain state.

However, in my opinion, the current version of the manuscript requires major improvement. Major issues involve interpretation and presentation of data.

First, linear filters were obtained from 204 S2 neurons out of 1157 recorded. No explanation is given as to why this radical weeding out of the data set. Presumably, the other >900 S2 neurons did not produce statistically significant filters. If so, this in itself is an important point. It means most neurons in the data set (over 80%) could not be characterized as being significantly responsive to these stimuli. In comparison, a majority of S1 neurons (54%) did yield filters. This is probably the major difference uncovered between both areas, because it suggests that the areas either (a) code stimuli with different degrees of sparseness or (b) are not equally captured by the paradigm used here, i.e. the majority of S2 neurons play roles not adequately explored by this analysis. Thus, statements like “in wS1, twice as many cells in the population encode a precise temporal pattern of whisker movement”, which are based on the number of neurons that actually were successfully captured by the analysis, in fact severely underestimate the extent to which the S2 subpopulation encoding precise whisker movement is a minority in the data. I am a big supporter of the feature selectivity framework for explaining neuronal responses, but on the basis of this dataset and the scarce information given, I’m afraid I can only be sceptical about its applicability (at least with these stimulation paradigms) to many S2 neurons. This crucial aspect (or issue) gets remarkably little discussion by the authors.

Second, some of the differences between areas that the authors do detect are best described as quantitative, and the paper somewhat unsatisfyingly distorts interpretation. A difference of 4-6 ms in temporal integration windows does not seem to be a qualitative step change in how the two areas behave; if the authors have evidence or reason to believe this is a fundamental difference, they don't provide it in the manuscript. So, while neglecting perhaps the biggest difference between the areas, which concerns the framework's applicability (previous paragraph), the authors oversell relatively minor differences between areas which however do happen to be captured by the framework. This therefore distorts the message on what S2 does and how it is different from S1.

Third, the rationale for the new PS-STA measure is unclear. I understand that it may be interesting to visualize how the stimulus projection onto a neuron's feature phase plane evolves over time, in the so-called temporal projection method. It might even be useful to show how these trajectories roam the phase spaces depicted in Fig. 2 thus determining spike probability. However, I do not see (and the manuscript doesn't really explain) why it is useful to introduce a new measure consisting of a spike-triggered average of the projection onto a spike-triggered manifold. What does this construction mean mathematically? What are its properties, are they sensible, and what do they mean for our biological understanding? The measure is introduced in an ad hoc manner, beyond the fact that its waveform is different for the S1 and S2 examples, and its meaning is never made clear. Similarly, what do the principal components in PS-STA shape in Fig. 5b-c correspond to? What do these differences in S1 and S2 density mean? Beyond learning that there is something different in S1 and S2 feature selectivity, one comes out of this analysis without clarity as to what these differences might mean in function. I suggest that without such a clearer picture, these differences don't much illuminate the reader, and the manuscript would be better off without them: they introduce complication and generate no insight.

Minor comments:

Phase-tuned neurons "were tuned to a small phase region" (p. 6). But the two examples (Fig. 2a and b) show that preferred phase regions can differ greatly in size. Are these differences in size less important than e.g. the number of peaks in the circular distribution of preferred phases?

Why were no supragranular neurons included in the data set? There almost seems to be a conscious decision not to have them.

Typos:

p. 7, Fig. 1b-c should be Fig. 1d-e

ref. 18 is incorrect

Fig. 6a-b, heatmaps are opaque and difficult to make sense of: they need proper labeling

The legend for Supplementary Fig. 4 does not properly describe the figure that's actually in the files

In reference to the manuscript:
NCOMMS-18-09372-T

Title: "Rich spatio-temporal stimulus dynamics unveil sensory specialization in cortical area S2"
Authors: M.A. Goldin*, E.R. Harrell*, L. Estebanez, D.E. Shulz

We have fully addressed all the Reviewer's comments. In summary, we made the following changes to the manuscript:

- (1) Clarified the fact that wS1 was measured in a previous study as requested by Reviewer #1
- (2) Improved the figure and legend description of Fig.1 as suggested by Reviewer #1
- (3) Elaborated on the conceptual advantages of the Temporal Projection Method (TPM) as suggested by Reviewer #1 and Reviewer #2
- (4) Made the introduction less antagonistic and included more citations as suggested by Reviewer #1
- (5) Discussed the quantity of neurons with significant TPM responses for the different types of stimulation and clarified why the spike-triggered covariance (STC) method is not sensitive enough in some cases, to respond to Reviewer #2's concerns about the lack of responsive cells in wS2
- (6) Added two Supplementary Figures to support our explanation about wS2 having a lower signal to noise ratio (SNR) than wS1
- (7) Explained with more detail the principal components of Fig. 5 for Reviewer #2
- (8) Clarified why we measured only in infragranular and granular layers, in response to Reviewer #2
- (9) Updated Fig. 2 and corresponding text due to an error in the original manuscript, thanks to the question raised by Reviewer #2 about the phase tuning
- (10) Corrected the typos pointed out by Reviewer #2
- (11) Updated numbering of references and Supplementary Figures to be cited in order

In what follows, we respond to each Reviewer's requests point by point:

Response to Reviewer #1

We are very grateful for the constructive criticism and helpful suggestions.

(1) *"The authors record from an impressively sample size of single units in wS2 (>1000). Although the Methods section describes wS1 recordings, the Results make it sound like wS1 results were from reanalysis of data recorded for an earlier study (please clarify!) They compare wS2 with wS1 results."*

Response: Indeed, the wS1 recordings were done in the same exact experimental conditions (strain, gender, age, weight, anesthetic, etc) and some of the data was reported in Estebanez et al. 2012. This is now clearly mentioned in the Results section and in the Methods section in these sentences:

"To compare, we applied the same analysis to 1038 neurons recorded previously in wS1¹⁷, which yielded 1320 significant linear filters from 567 responsive neurons."

"wS1 experiments were already reported in a previous work¹⁵ and here we perform new analyses on the data obtained."

(2) *"I found Figure 1 hard to understand. This may be improved by adding more relevant detail to the legend."*

Response: To address the Reviewer's concern we have improved the figure and the legend. The lower part of panel (a) in the figure is now more descriptive and this part of the legend reads:

"(a) Schematic of the recording setup. Top: The right 24 caudal macrovibrissae of an isoflurane-anesthetized rat are trimmed and placed inside plastic tips attached to piezo-electric benders. Only two stimulators are shown, each controlling the whisker movement along the rostral-caudal (R-C) axis. All

whiskers received the same Gaussian white noise deflections (in position). Multi-shank silicon probes are placed in either wS2 or wS1 to record neuronal responses. Bottom: The spike-triggered covariance (STC) technique involves tabulating the stimulus shapes preceding each spike from a single unit. This table of spike-eliciting stimuli is called the spike-triggered ensemble (STE). A covariance based, PCA-like analysis is then carried out on the STE to recover the stimulus shapes, or filters, most correlated with spiking. The same process is then applied to shuffled spike trains to determine which filters are significant (See Methods).”

In the rest of the legend, we added references for abbreviations and color details that appear in the figure, and fixed formatting issues (some of the panel letters were not in bold).

(3) *“Regarding the authors’ temporal projection method (TPM, depicted in Figure 3b): I get that, as they state, the “...procedure is a transformation from whisker position coordinates to a new coordinate system which is empirically determined to be most relevant for the neurons.” But I wasn’t clear about its underlying logic.”*

Response: Classical reverse correlation analyses require an extensive amount of data. Indeed, the first step of this analysis is to retrieve filters (i.e. the stimulus shape or shapes with the highest probability of eliciting a spike). This process requires increasing amounts of data as the dimensionality of the stimulus increases. Even with our limited 50 ms stimulus window sampled every 2 ms, there are 25 temporal dimensions in a spike-eliciting whisker movement. If we consider a necessary minimum of 50 spikes per dimension (heuristic given in Rust et al. 2005 and used in Estebanez et al. 2012), we need around 1250 spikes if all whiskers are moved identically (fully correlated stimulation). This number increases to 30000 spikes when all 24 whiskers receive independent deflections (uncorrelated stimulations). Given that the basal firing rates in the two cortical areas we study are low (~ 0.5 Hz, Supplementary Fig. 5), it would take marathon recordings to obtain anywhere near these spike counts for a large population of neurons.

This leads to a necessary compromise in the case of data with large dimensionality such as the multi-whisker, high-frequency stimulus that we apply to the whiskers.

Traditionally, this issue has been solved by reducing the time range around the spikes where the stimulus is extracted, as well as the spatial/temporal resolution of the stimulus (how many whiskers to consider and if time can be down-sampled). Here, we have devised the temporal projection method as an alternative solution that avoids these information losses. To do so, we base our analysis on the observation that a shared 2D feature space – previously identified in wS1 (Estebanez et al. 2012) and once again apparent in wS2 – appears to be a fundamental aspect of whisker cortical responses to white noise stimulation. On this basis we apply a dimensionality reduction of the stimulus by projecting it into this subspace, and we further reduce dimensionality by focusing on the amplitude (and not the phase) of the stimulus projection into the shared space. We are thus directing our study to empirically defined dimensions of interest for the neurons.

Thanks to this simple feature-based dimensionality reduction step, we can:

- (1) Increase the signal to noise ratio (SNR) of the analysis by focusing on the subspace of the stimulus that is defined by the dominant pair of filters identified by classical reverse correlation (STC). This amounts to including many more neurons in our study, as neurons that did not fire enough or have a high enough SNR to give significant filters can give significant TPM PS-STAs.
- (2) Extend the time window that is included in the analysis from a 50 ms range for classical STC to a 150 ms interval. This reveals previously unreported temporal dynamics in the spike-eliciting whisker movements.

The improvement of the SNR due to this approach leads to a marked increase in the number of neurons displaying functional responses, with the numbers of responding cells detailed in the table below. In terms of total cells showing a response to either correlated or uncorrelated stimulation, using TPM we found 943 responsive cells in wS2 and 942 in wS1 ($943/1157 = 81\%$ for wS2 and $942/1038=91\%$ for wS1).

	wS1 correlated	wS2 correlated	wS1 uncorrelated	wS2 uncorrelated
# neurons STC	567	204	680	5
# neurons TPM	828	679	870	735

We have updated the manuscript in many places to add these numbers and to clarify the logic behind the TPM method.

Here is the text summarizing our response found on page 9-10:

“The use of TPM with continuous whisker movements uncovers extended stimulus dependencies for many neurons that yield no significant filters using classical STC. In wS2, we find 679 neurons with significant PS-STAs, while we were only able to recover STC filters for 204 of these cells. For wS1, we detect 828 PS-STAs from the same population of 1038 neurons that yielded filters for 567 cells. TPM is much less data hungry than the classical STC analysis, and it can partially compensate for the fact that wS2 responses, although present, seem to be less strong than wS1 responses in terms of spikes/filter-like stimulus (Supplementary Figs. 1 and 3). The reason why the TPM is more sensitive at detecting functional responses than classical STC is that once the stimulus subspace of interest for the cells in a region is known, we can focus our analysis in this subspace, thus increasing the signal-to-noise ratio on which we base our functional assessment. This added sensitivity, coupled with its applicability across longer time windows than STC make it a useful tool to obtain rich dynamic stimulus dependencies.”

In the Results section, we report on the TPM numbers for uncorrelated stimulus:

“As with the correlated stimulation pattern, the TPM greatly increased the number of detected functional responses during uncorrelated stimulation. We detected 735 responsive neurons from wS2 (vs 5 with STC) and 870 from wS1 (vs 680 with STC).”

In the Methods section, we report on the total responses:

“We found that 943 and 942 neurons were responding either to the correlated or the uncorrelated stimulation with our method in wS2 and wS1 respectively. This supports its general applicability and capacity to capture responses in the whisker system under complex stimulation conditions, especially in the wS2 region which more sparsely encodes longer and spatially spread stimulus features.”

(4) *“In general, I think the paper is going to be very hard to understand for readers not already familiar with spike-triggered covariance analysis. Some effort to make it more accessible would be appreciated by many readers.”*

Response: To address this concern, we have both improved Figure 1 and its legend, as well as added this brief explanation at the beginning of the Results section:

“From the spiking activity observed during these correlated stimulations, we applied spike-triggered covariance (STC) analysis. Briefly, this method entails tabulating the whisker movement shapes preceding each spike from a single unit, computing the covariance matrix of these whisker movements, and using eigenvector decomposition of this matrix to find the whisker movements, or filters, responsible for eliciting the most spikes (See Methods for details).”

(5) “Many anatomical tracing studies agree that based on the thalamic input structure and the nature of their reciprocal connectivity, wS2 and wS1 can be thought of as equally placed in the somatosensory hierarchy^{8,9}.” This is hardly a consensus view. There is a long history across multiple species of addressing this question. My own view is that S2 is more integrative and based on multiple lines of evidence situated at a higher level of the cortical hierarchy. To take one feature, wS2 sends a dense projection to layer 1 of wS1, whereas wS1-to-wS2 projections do not specifically target layer 1. For some recent treatment of this issue in limb cortex, see:

Suter, B.A., and Shepherd, G.M. (2015). Reciprocal interareal connections to corticospinal neurons in mouse M1 and S2. *J Neurosci* 35, 2959-2974.

Response: We have updated the introduction to reflect this lack of a consensus. This paragraph now reads:

“Many anatomical tracing studies suggest that based on the thalamic input structure and the nature of their reciprocal connectivity, wS2 and wS1 can be thought of as equally placed in the somatosensory hierarchy⁸⁻¹¹. Direct thalamic input to wS2 comes through both the extralemniscal and paralemniscal pathways¹², while corticocortical projections from S1 to S2 originate in layers 2, 3, and 5a of S1 and terminate in extragranular layers of S2. The reciprocal connections from S2 to S1 follow the same connection pattern⁹. In the mouse, infragranular cells in wS1 receive more numerous connections from infragranular wS2 cells, and likewise supragranular wS1 cells receive more connections from supragranular wS2 cells¹¹. Although there is still some controversy, these data suggest that sensory processing at the level of wS2 and wS1 in rodents is done in an interdependent, parallelized manner.”

Regarding the layer 1 specificity that is mentioned by the Reviewer from the Suter and Shepherd report, this reference gives a detailed electrophysiological account of the connections in the forepaw regions of M1 and S2, which can be compared with the connections found between S1 and M1 (Mao et al. 2011). Our views on the direct connections between S1 and S2 are supported by the following articles, which are all now included in our references. The last two references (iConnectome and Denardo et. al) were not present before and have been added to the manuscript.

- Chakrabarti, S. & Alloway, K. D. Differential origin of projections from SI barrel cortex to the whisker representations in SII and MI. *J. Comp. Neurol.* (2006). doi:10.1002/cne.21052
- Koralek, K.A., Olavarria, J. and Kellackey, H.P., Areal and laminar organization of corticocortical projections in the rat somatosensory cortex. *Journal of Comparative Neurology*, 299(2), pp.133-150. (1990).
- Liao, C.-C. & Yen, C.-T. Functional connectivity of the secondary somatosensory cortex of the rat. *Anat. Rec. (Hoboken)*. **291**, 960–973 (2008).
- the Mouse iConnectome Project, to be cited as “Zingg, B., Hintiryan, H., Gou, L., Song, M.Y., Bay, M., Bienkowski, M.S., Foster, N.N., Yamashita, S., Bowman, I., Toga, A.W. and Dong, H.W., 2014. Neural networks of the mouse neocortex. *Cell*, 156(5), pp.1096-1111.”
- DeNardo, L.A., Berns, D.S., DeLoach, K. and Luo, L., 2015. Connectivity of mouse somatosensory and prefrontal cortex examined with trans-synaptic tracing. *Nature neuroscience*, 18(11), p.1687.

(6) *The reciprocal connections from wS2 to wS1 are symmetric. What do the authors mean by symmetric? If they mean “homotypic” (somatotopically matched areas connecting to one another), they are definitely lacking appropriate citations (also, the Liao and Chen study cited deals with paw and not whisker areas).”*

We removed the word symmetric and replaced it with a more appropriate description. We also removed the ‘w’ from our abbreviations when studies were not focused on whisker regions of the somatosensory cortical areas. Now this part in the Introduction reads:

“Direct thalamic input to wS2 comes through both the extralemniscal and paralemniscal pathways¹², while corticocortical projections from S1 to S2 originate in layers 2, 3, and 5a of S1 and terminate in extragranular layers of S2. The reciprocal connections from S2 to S1 follow the same connection pattern⁹.”

(7) *“Before making any assumptions about the roles of wS2 and wS1 in perception, it is important to adequately assess wS2’s sensory representation.” I agree that assessing wS2’s sensory representation is an important goal. But it is unfair to imply that the studies cited [11,12] made “assumptions” about the roles of wS2 and wS1---rather, the roles of these areas were experimentally investigated, not assumed. Moreover, it is far from clear that wS2’s sensory representation can be “adequately” assessed under anesthesia, and with only rostral-caudal whisker deflections, as in the current study. I suggest that the tone of the entire introduction up to this sentence could be softened---the claims are overly strong and not adequately cited.”*

Response: We agree with the Reviewer. Our tone was unintentionally too antagonistic. We have attempted to soften our tone and focus more on our approach using multi-whisker stimulation. Our updated manuscript reads:

“While there is anatomical evidence for parallel processing of somatosensory information in wS2 and wS1, recent perceptual studies in mice have highlighted wS2’s role in the choice-related, top-down flow of information^{13,14}. Although choice-related activity in wS2 is more predominant than it is in wS1, it is unknown whether the single whisker periodic deflection used in these studies adequately engages wS2’s sensory representation. With such a stimulus, there is no global or multi-whisker component, which could be an important factor in delineating wS2’s sensory function^{6,15}. With this in mind, we set out to assess wS2’s sensory representation during multi-whisker stimulation. To this end, the first important question to ask is what patterns of whisker movement are salient for wS2 neurons, and how these whisker movement patterns differ from those represented in wS1. Accordingly, this work aims to provide a detailed characterization of the responses of large populations of single units recorded in wS2 during multi-whisker stimulation.”

Response to Reviewer #2

We want to thank the Reviewer for insightful constructive criticism. Here, we respond to all remarks:

(1a) *“First, linear filters were obtained from 204 S2 neurons out of 1157 recorded. No explanation is given as to why this radical weeding out of the data set. Presumably, the other >900 S2 neurons did not produce statistically significant filters. If so, this in itself is an important point. It means most neurons in the data set (over 80%) could not be characterized as being significantly responsive to these stimuli. In comparison, a majority of S1 neurons (54%) did yield filters. This is probably the major difference uncovered between both areas, because it suggests that the areas either (a) code stimuli with different degrees of sparseness or (b) are not equally captured by the paradigm used here, i.e. the majority of S2 neurons play roles not adequately explored by this analysis.”*

Response: We thank the Reviewer for making this observation, which we did not adequately comment on in our initial manuscript. First, we will address the Reviewer to our response to remark 3 from Reviewer #1, which describes the amount of data required to retrieve filters using classical STC. Indeed, there are fewer neurons that give significant filters in wS2, so the problems described above are exaggerated in this region, and we think this is due to a reduced signal-to-noise ratio (SNR) in the wS2 firing activity (perhaps the same as option (a) of the Reviewer’s remark). It has already been

reported that there are fewer spikes/stimulus in wS2 (Kwegyur et al. 2004), so this means lower ‘signal’, and on top of that there is more jitter in the response measured during both sparse noise stimulation and Gaussian white noise stimulation, which is clear in our data in Figure 4 for both the single cells (see PS-STA and histogram spread in Figure 4a-b upper and lower panels respectively) and across the population (Figure 4c-d). This jitter can also be understood as the C-width in Supplementary Fig. 4, which is much larger in wS2. As further evidence for this, we have added two supplemental figures. The first one (Supp. Fig. 1b) contains a 1-D representation of all the non-linear functions (NLFs) for phase-invariant cells from wS1 and wS2. These NLFs are much more sloped at the edges in wS1, meaning that cells fire more for a filter-like stimulus. The second (Supp. Fig. 3a-b) shows the averages of the significant PS-STAs obtained during correlated stimulation in wS1 and wS2, both absolute averages and normalized. These averages show that wS1 has a much sharper, higher average peak, whereas for wS2 the peak is lower and wider. If we accept that wS2 responses have a lower SNR or code more sparsely, we would expect to obtain fewer filters for a fixed significance level, which is necessary to be consistent across the two regions. This is part of the reason for introducing the TPM in this study, to see if there are fewer filters because the cells do not respond to the stimulus (this would also be visible from a difference in spiking activity, which we do not observe, see Supp. Fig. 5) or if it is just a matter of SNR/sparseness. We think that the fact that across all stimulations we find 943 responsive cells in wS2 and 942 responsive neurons in wS1 using the TPM shows that both areas respond to the stimulus, albeit with different properties and reliabilities. The PS-STA shape, which we will return to for the response to remark 4, captures these differences (i.e. spread can be thought of as the reliability or jitter of the spike timing after a spike-eliciting stimulus, like a stimulus resembling a region-specific filter). We added the following text to the first paragraph in the Results section to make sure this issue does not seem overlooked:

“Thus, there is a major difference in the number of cells yielding significant filters in the two regions (204/1156 in wS2 vs. 567/1038 in wS1). This observation could be due to a real difference between the whisker movements coded in the two regions (with wS1 more strongly activated by features contained in globally correlated white noise) or reflect a difference in the signal-to-noise-ratio (SNR) of the responses in the two regions (where less spikes/stimulus in wS2 results in lower filter yields).”

(1b) Thus, statements like “in wS1, twice as many cells in the population encode a precise temporal pattern of whisker movement”, which are based on the number of neurons that actually were successfully captured by the analysis, in fact severely underestimate the extent to which the S2 subpopulation encoding precise whisker movement is a minority in the data. I am a big supporter of the feature selectivity framework for explaining neuronal responses, but on the basis of this dataset and the scarce information given, I’m afraid I can only be sceptical about its applicability (at least with these stimulation paradigms) to many S2 neurons. This crucial aspect (or issue) gets remarkably little discussion by the authors.”

Response: We fully agree that the STC framework selects the highly responsive cells and that by looking only at this subpopulation of high responders, we are probably overestimating how much encoding of precise whisker movements exist across the population in wS2 (31/204 and 31/1157 is a huge difference). We still think that an established upper bound for the percentage of the population encoding precise whisker movements is useful, and even if the same ‘weeding out’ process does not have as strong an effect in wS1, it is still a noteworthy observation. To be able to compare on a more level playing field, we developed the TPM. All the new counts for responsive neurons are now reported in the manuscript as mentioned in the previous remark. With this method we were able characterize the differences between wS2 and wS1 by comparing the PS-STA for a much larger number of responsive cells than was possible with STC filter-generating cells.

(2) “Second, some of the differences between areas that the authors do detect are best described as quantitative, and the paper somewhat unsatisfyingly distorts interpretation. A difference of 4-6 ms in

temporal integration windows does not seem to be a qualitative step change in how the two areas behave; if the authors have evidence or reason to believe this is a fundamental difference, they don't provide it in the manuscript."

Response: We agree that a 4-6 ms difference may seem small, but we believe that it can still be important. First, it is a difference in half-width across the population, not an absolute difference of 4-6 ms. Second, this difference is echoed by many other pieces of data in the paper. The PS-STAs in wS2 are wider as can be seen in the example cell in Fig. 3, and the population average we have now added in Supplementary Fig. 3. Also, in the classical sparse noise stimulation, the histograms from the wS2 cell in Fig. 4 are wider (the spikes from a single cell after a single whisker ramp stimulus are not as well-locked to the stimulus as in wS1), and there is more variation in the latencies across the population. All these facts lead us to believe that this temporal element is an important coding difference between the two areas. We interpret in our Discussion section that this temporal imprecision, or extended temporal integration, is not a bug but in fact a feature. To compute global statistics from a tactile scene, not only the spatial extension must be broader (more whiskers), but also increased temporal integration can facilitate the gathering of stimulus information from many different whiskers in an efficient manner. The intuition is that if the whisker inputs to a single neuron are temporally very sharp as in wS1, the coding space would explode in terms of multi-whisker spatio-temporal combinations. The wS2 neurons thus summate inputs over longer temporal windows because they are less interested in precise single whisker events and more interested in multi-whisker synergies.

We added this sentence to the Discussion section to more directly state that we think the temporal differences are important:

"If the single whisker contributions to the firing of a single neuron are temporally very sharp as in wS1, the coding space would explode in terms of multi-whisker spatio-temporal combinations. The wS2 neurons sum inputs over longer temporal windows to more efficiently encode multi-whisker synergies."

(3a) *"Third, the rationale for the new PS-STA measure is unclear. I understand that it may be interesting to visualize how the stimulus projection onto a neuron's feature phase plane evolves over time, in the so-called temporal projection method. It might even be useful to show how these trajectories roam the phase spaces depicted in Fig. 2 thus determining spike probability. However, I do not see (and the manuscript doesn't really explain) why it is useful to introduce a new measure consisting of a spike-triggered average of the projection onto a spike-triggered manifold."*

Response: We kindly direct the Reviewer to our response to Reviewer #1 on remark 3, which we think is a good start to answer this question/concern. Taking it one step further, when we make a normal PSTH during sparse noise stimulation, we align all the spikes with respect to the onset of a fixed stimulus, a whisker ramp for example, and we analyze the features about where these spikes fall with respect to the fixed stimulus (latency, peak, sharpness, etc). A PS-STA inverts this idea. We align a depiction of the stimulus (how 'filter-like' is the stimulus that occurred before a spike) with respect to the spike times. When we average across all spikes, we can get an idea, compared to the average stimulus, how different is the average **spike-eliciting** stimulus in terms of 'filter-like-ness'. This quantification can be extended backwards in time well before the spikes occurred and gives an interesting picture of the stimulus dynamics that occur before spikes.

We have added this intuitive explanation to our Methods section and updated the text in these two positions to clarify what the PS-STA brings:

"Polar coordinates are more convenient because the spiking activity for neurons in both wS2 and wS1 occurs predominantly at high values of the radial coordinate, which correspond to relevant filter-like whisker movements, as shown in Fig. 2"

“The use of TPM with continuous whisker movements uncovers extended stimulus dependencies for many neurons that yield no significant filters using classical STC. In wS2, we find 679 neurons with significant PS-STAs, while we were only able to recover STC filters for 204 of these cells. For wS1, we detect 828 PS-STAs from the same population of 1038 neurons that yielded filters for 567 cells. TPM is much less data hungry than the classical STC analysis, and it can partially compensate for the fact that wS2 responses, although present, seem to be less strong than wS1 responses in terms of spikes/filter-like stimulus (Supplementary Figs. 1 and 3). The reason why the TPM is more sensitive at detecting functional responses than classical STC is that once the stimulus subspace of interest for the cells in a region is known, we can focus our analysis in this subspace, thus increasing the signal-to-noise ratio on which we base our functional assessment. This added sensitivity, coupled with its applicability across longer time windows than STC make it a useful tool to obtain rich dynamic stimulus dependencies.”

(3b) What does this construction mean mathematically? What are its properties, are they sensible, and what do they mean for our biological understanding? The measure is introduced in an ad hoc manner, beyond the fact that its waveform is different for the S1 and S2 examples, and its meaning is never made clear.

Response: This description has been added to the methods:

“The TPM is mathematically very similar to standard linear-nonlinear Poisson models, but with important extensions. The stimulus first passes through a filter bank of two filters (products of this linear operation are x and y), then computing the radial coordinate can be thought of as the nonlinear part ($\sqrt{x^2+y^2}$) of LNP. This can be interpreted either as a ‘subspace energy’ like from the Adelson energy model, or as just a vector strength. The output of this process (the radius) is then averaged across times leading up to and following all spikes (-150ms to +50ms). This visualization helps to see that the average radius just before spiking is higher than chance (as would be expected for neurons giving filters because this is the standard LNP), but also that is not the complete story of what causes the neuron to fire. Extending this windowing backward shows a more elaborate stimulus dependence, which varies between wS2 and wS1. Biologically, this can be interpreted to mean that when we shake the whiskers vigorously or the animal shakes them as such, there is not just a simple fixed feature detection going on, the dynamics are extended and depend on stimulus history at longer latencies.”

(3c) Similarly, what do the principal components in PS-STA shape in Fig. 5b-c correspond to? What do these differences in S1 and S2 density mean? Beyond learning that there is something different in S1 and S2 feature selectivity, one comes out of this analysis without clarity as to what these differences might mean in function. I suggest that without such a clearer picture, these differences don't much illuminate the reader, and the manuscript would be better off without them: they introduce complication and generate no insight.”

Principal components PC1 and PC2 are described in detail in Supplementary Figure 4. There were some problems with the uploading of this figure so perhaps some information was lost in the process. We now include a description at the end of the Results subsection:

“The separation between wS2 and wS1 lies mainly along the first principal component which is derived from the spread, latency, and sharpness of the evoked responses in both correlated and uncorrelated stimulation patterns (magenta bars in Supplementary Fig. 4n). The second principal component is related to the ratios between responses to the correlated and uncorrelated stimulation (cyan bars in Supplementary Fig 3n). The bigger spread in the second principal component of wS2 compared to wS1 emphasizes the diversity of the different responses across the two stimulation patterns.”

We consider that this image is important for the paper. Some interpretations can be drawn by looking into the individual feature distributions of the PS-STAs given in Supplementary Fig. 4. But this PCA-based summary is illustrative of how we conceptualize wS2 and wS1 working together and represents just how much total separation there is across all quantified domains. The cortical regions receive some of the same thalamic inputs and are highly interconnected. Therefore, to expect them to have completely unique, separable functions is probably incorrect. The point of showing these regions and cell densities is to illustrate that while there are specialized parts of the coding schemes in these two areas, corresponding to the areas with much more density from one of the regions, there are also shared parts of the coding space where the densities are close to equal. Different cell densities in different regions of the coding space imply that higher percentages of the respective populations occupy certain functional regions. The conclusion is functional specialization with some overlap.

This is mentioned in the Results now:

“Different cell densities in different regions of the coding space imply that higher percentages of the respective populations occupy certain functional regimes.”

This is also mentioned in the Discussion:

“Within these regions, there is also redundancy in the representations, which is not surprising considering they are highly interconnected and share thalamic inputs.”

(4) *“Phase-tuned neurons “were tuned to a small phase region” (p. 6). But the two examples (Fig. 2a and b) show that preferred phase regions can differ greatly in size. Are these differences in size less important than e.g. the number of peaks in the circular distribution of preferred phases?”*

Response: Yes, we did not report on the spread of the tuned regions. We have carried out this analysis and there are significant differences which we have now added into Supplementary Fig. 1. The average tuning half-width of the wS2 and wS1 populations are 80 and 64 degrees respectively (population mean of half-width half-maximum value of tuning peak). This difference is significant ($p < 0.05$, permuted cell-region mappings), and it represents half a bin in our angular representation (our angular bins are 30 degrees). We think that these facts taken together, less phase-selective neurons and larger phase regions for those that are phase selective, adequately support our conclusion that wS2 is less concerned with fine tactile features.

(5) *“Why were no supragranular neurons included in the data set? There almost seems to be a conscious decision not to have them.”*

Response: There was no purposeful decision to leave them out, but there are two main reasons for that. First, to compare with wS1 previously reported data (Estebanez et al. 2012) we needed to aim at the same layers they recorded. Second, since we wanted to make reverse correlation analyses of large populations of cells using multi-electrodes, we needed to target cell-dense regions with adequate spiking activity, which is not the case for supragranular layers. We added this information at two places in the Methods:

“To obtain sufficient spiking activity, all recorded neurons from wS2 were in granular or infragranular layers (layers 4, 5, and 6)”

“The reason to aim at granular and infragranular layers relies on the need for sufficient spiking activity to carry out STC analysis (see Data analysis below)”

(6) *“p. 7, Fig. 1b-c should be Fig. 1D-e; ref. 18 is incorrect; Fig. 6a-b, heatmaps are opaque and difficult to make sense of: they need proper labeling; The legend for Supplementary Fig. 4 does not properly describe the figure that’s actually in the files”*

Response: We thank the Reviewer for his/her advice to make the figure more legible and improve other errors in the manuscript. To correct these errors, we made the following changes:

- page 5, we changed twice “(Fig. 1c-d)” for “(Fig.1d-e)”

- reference was incorrect, we put now the correct item in the references: “Jacquin, M.F., Renehan, W.E., Rhoades, R.W. and Panneton, W.M. Morphology and topography of identified primary afferents in trigeminal subnuclei principalis and oralis. J. Neurophysiol. 70, 1911-36 (1993)”

- Figure 6a-b, we changed the contrast for the heatmaps, added the same labeling as in Fig 4a. for the responding whiskers and a grid to ease the view. We also modified the text into the legend. Now the modified part reads: “Left Top: whisker pad heatmap (same arrangement as in Fig 4a)”

- we downloaded the Supp. Fig. 4 pdf file from the on-line submission system and it seems to correspond with our legend in the text file. It could be possible that there was a technical issue here, since this file had to be manually converted by an Editorial Assistant, Sean Speers, after the system was unable to convert it. Our correspondence with him dates from April 4th. We are now providing a single pdf with the Supplementary Figures and captions as is required.

Reviewers' comments:

Reviewer #1 (Remarks to the Author):

I thank the authors for their thoughtful and thorough response to my earlier review, and in particular for including in their rebuttal letter a detailed and helpful explanation of rationale behind the TPM method. I think the manuscript is now in strong shape.

Reviewer #2 (Remarks to the Author):

The authors have worked hard to address our concerns and have answered my comments to some extent. The approach involved in their TPM/PS-STA method is now partly clearer. However, there is still an important aspect in which I still do not understand what is going on. The manuscript states, "first project the raw stimulus (RS) into the relevant subspace. This is done by computing the dot product of a 50ms stimulus window with each relevant filter, which gives two coordinates." Is the "relevant subspace" the one computed for wS1 or wS2 as a whole? If so, why should a particular neuron with no significant STC-derived filters have responses related to the relevant subspace? Or is the projection done onto a particular subspace computed for that specific neuron? I don't see the logic that justifies taking a neuron *outside* of the original set of cells well captured by STC (e.g. 204 for wS2) and projecting the stimuli that evoked spikes in that neuron onto a generic "relevant subspace" constructed for a different set of neurons. Why is the method expected to work? What are its foundations? Why should we trust its output for those neurons *not* originally captured by STC? This is related to my earlier comments in which I stated that I didn't understand the rationale of this method. While the authors have clarified the procedure, they have not clarified the logic of the idea. I am not asking for a level of mathematical rigor comparable to what others have provided by STA/STC analyses (e.g. Paninski et al) but I would like a better understanding of why the method is expected to provide insight into the functional properties of those neurons.

Response to Reviewer #2

Reviewer: "The authors have worked hard to address our concerns and have answered my comments to some extent. The approach involved in their TPM/PS-STA method is now partly clearer. However, there is still an important aspect in which I still do not understand what is going on. The manuscript states, "first project the raw stimulus (RS) into the relevant subspace. This is done by computing the dot product of a 50ms stimulus window with each relevant filter, which gives two coordinates." Is the "relevant subspace" the one computed for wS1 or wS2 as a whole?"

Response: The "relevant subspaces" are the two-dimensional spaces derived from STC analysis on the respective populations of neurons in wS1 and wS2. There is a relevant filter space for wS2 (generated by the filters shown in Figure 1b in red and blue solid lines) and a relevant filter space for wS1 (generated by the filters shown in red and blue dashed lines in Figure 1c). These spaces are derived from the underlying populations of significant filters in the respective regions. They do not belong to any single neuron, but instead provide a compact representation that can describe almost all of the single neuron filters. This is shown by the proximity to the unit circle of almost every projected significant filter encountered (Figure 1d for wS2 and Figure 1e for wS1).

Reviewer: "If so, why should a particular neuron with no significant STC-derived filters have responses related to the relevant subspace?"

Response: We think that the answer to this question is one of the important results of our paper, which we hope to address to the satisfaction of the Reviewer.

There are two main reasons why neurons that do not give filters still respond to stimuli that have high projections into the relevant subspace.

1) **Statistical power** – The mean firing rates in our recording conditions are about ~1.5 Hz in both wS2 and wS1 (Sup. Figure 5). The recordings last 2.5 hours, but the protocols alternate between correlated noise, uncorrelated noise, and sparse noise ramps in a randomized manner. In 2.5 hours, we get 1 hour of spikes during correlated stimulation and 1 hour of spikes for uncorrelated stimulation (~ 5400 spikes for the average cell). While the heuristic of 50 spikes per dimension of stimulus (*Rust, N.C., Schwartz, O., Movshon, J.A. & Simoncelli, E.P. Spatiotemporal elements of macaque v1 receptive fields. Neuron 46, 945–956 (2005)*) provides a general idea of how many spikes are needed to find significant filters, it is probably much more than that if we want to find all the dimensions of interest for a cell. For us, the statistical cutoff to be deemed a significant filter was that the eigenvalues had to be **8 standard deviations** above the shuffled distribution, which is standard practice in the field (Estebanez et al. 2012, Rust et al. 2005). For many cells, this stringent cutoff eliminates them despite having responses above the shuffled distribution (although not the full 8 standard deviations). These stringent cutoffs are used in the field to be able to apply consistent thresholds across large populations of neurons with different levels of spiking activity. The result is that many cells are thrown out but **DO** have tuning in the relevant subspace. They could be discarded due to not having enough data (either did not fire enough or we did not hold them for the full recording) or because their functional responses are sparse. The

statistical question that our TPM method asks is “Does this cell have more spikes coming from filter-like stimuli than would be expected if it did not care about filter-like stimuli?” The answer is a resounding yes. Many cells fire more for filter-like stimuli than would be expected if they were not concerned at all with the relevant subspace. This can be seen when looking at the projection into the relevant subspace of the top eigenvectors for cells that did not make the 8-sigma cutoff but do have significant PS-STA responses (Supp Fig 3g). This is one of many new panels we have added to Supplementary Figure 3. In Sup. Fig 3c and 3e, we divided the PS-STA traces into three groups: the STC and TPM significant (STC+ TPM+, blue for 3c and red for 3e), the STC not significant but TPM significant (STC - TPM +, cyan for 3c and magenta for 3e), and the cells that did not give a response for either type of analysis (STC- TPM-, orange in both 3c and 3e). We see that for both wS2 and wS1, STC- TPM+ cells have a mean PS-STA (based on z-score) value at time = 0 s below the 8-sigma cutoff. However, it is clear that these groups of cells still show the dynamics present in the STC+ group, as can be seen in the normalized traces in Sup. Fig 3d and 3f. On the contrary, STC-TPM- cells show no dynamics and no signal.

We also computed the firing rate of these groups, which supports our observation that low firing rates are the main reason that some cells fail to provide significant STC filters. These firing rates are provided in the following table:

	All cells (Hz)	STC+, TPM+ (Hz)	STC-, TPM+ (Hz)	STC-, TPM (Hz)
wS1	1.5	2.0	1.1	0.6
wS2	1.5	2.9	1.6	0.8

Firing rates are almost double for the cells that are STC+ compared to the STC- TPM+ both for wS2 and wS1.

As a further check of our result for wS2, which raised the Reviewer’s concerns, we show in Sup Fig 3g the top STC eigenvectors obtained for the group of cells that are STC- projected onto the relevant subspace. We can see that these eigenvectors are much closer to the unit circle than those for TPM- cells, meaning that the stimulus dimensions most strongly correlated with the spiking activity in the STC- TPM+ cells fall within the relevant subspace.

2) Shortcomings of traditional STC – Traditional STC makes assumptions about the functional activity of cells. In its traditional use, a ~50-60 ms window of pre-spike stimulus is examined and a set of features occurring in this 50-60 ms window is obtained. These ‘static’ features are then modeled and anything that happens before 60 ms is not considered relevant for the cell’s responses. The underlying logic of this is that the cell takes some kind of ‘snapshot’ of the stimulus and can either produce a spike or not, and all of the stimulus dependence falls within this short window of integration. We observe that the feature detection in our conditions is not a static snapshot. If we look within the relevant spaces long

before the spikes, the stimulus values are not random, in our case they have lower subspace projections than chance. This is now emphasized in Sup. Fig 3c and 3e, where we explicitly point out the snapshot that STC makes, whereas we can obtain all the dynamics present in the stimulus with our TPM. This requires more attention, since the dynamics become even richer in the uncorrelated stimulation with the addition of spatial dimensions. For that we now emphasize that TPM vs STC means ‘static-snapshot’ vs ‘dynamic-video’. This is another factor that makes the STC- TPM+ neurons have a lower chance of producing STC filters. As we see in Sup Fig 3d and 3f normalized PS-STAs, these groups of cells have a broader temporal integration compared with the STC+ group, again decreasing the statistical power to obtain significant STC filters.

Reviewer: “Or is the projection done onto a particular subspace computed for that specific neuron?”

Response: No, the projection is done on a region-wide space as described above.

*Reviewer: “I don't see the logic that justifies taking a neuron *outside* of the original set of cells well captured by STC (e.g. 204 for wS2) and projecting the stimuli that evoked spikes in that neuron onto a generic "relevant subspace" constructed for a different set of neurons. Why is the method expected to work? What are its foundations? Why should we trust its output for those neurons *not* originally captured by STC? This is related to my earlier comments in which I stated that I didn't understand the rationale of this method. While the authors have clarified the procedure, they have not clarified the logic of the idea. I am not asking for a level of mathematical rigor comparable to what others have provided by STA/STC analyses (e.g. Paninski et al) but I would like a better understanding of why the method is expected to provide insight into the functional properties of those neurons.”*

Response: We will tackle these remarks one at a time.

*“I don't see the logic that justifies taking a neuron *outside* of the original set of cells well captured by STC (e.g. 204 for wS2) and projecting the stimuli that evoked spikes in that neuron onto a generic "relevant subspace" constructed for a different set of neurons.”*

The only line separating STC filter-producing cells and the other cells is a stringent statistical line made using traditional STC methods. It is not meant to imply that there are two separate populations of neurons with potentially different functions (one relevant filter-responsive and the other not). We observed that if a cell did give filters, the filters fall into the subspace. If a cell did not give significant filters but did have significant responses when applying our new method (TPM+), the eigenvector with the highest eigenvalue generally still fell into the subspace (see new Sup. Fig 3g, left), suggesting they respond to the same kinds of stimuli above other stimulus dimensions, just not with a high enough signal to noise ratio to pass the statistical tests. All neurons not responding to our method show a poor projection into the relevant subspace (Fig 3g, middle). Both groups can be compared in Sup Fig 3g (right).

Our TPM asks a different question than traditional STC, which allows it to capture the non-STC-filter-producing cells because it is focused on the experimentally determined relevant dimensions, and thus avoids the curse of high dimensionality associated with STC.

“Why is the method expected to work?”

It is expected to work because we see that many cells are responding to relevant filter-like stimuli but below STC statistical thresholds. Statistically, it is more powerful since we can reduce the dimensionality by a factor of 25. Therefore, it is less computationally demanding, such that we could make 1000 shuffled distributions to compute our Z-score in a few hours, while for the STC method we had to rely on 200 shuffles that had to be run over several days for significance detection.

“What are its foundations?”

Its foundations are the same as those for the standard LNP models, but it makes a temporal extension. We developed it because we saw that many neurons were responding to stimuli that were relevant-filter like but did not give filters using classical STC. We tried to find novel ways to address the problems associated with high-dimensionality.

*“Why should we trust its output for those neurons *not* originally captured by STC? This is related to my earlier comments in which I stated that I didn't understand the rationale of this method. While the authors have clarified the procedure, they have not clarified the logic of the idea. I am not asking for a level of mathematical rigor comparable to what others have provided by STA/STC analyses (e.g. Paninski et al) but I would like a better understanding of why the method is expected to provide insight into the functional properties of those neurons.”*

We should trust its output for those not captured by STC because we have demonstrated in Sup Fig 3 that the reason STC- TPM+ cells are not captured by STC is mostly due to statistical limitations and insufficient spiking for high-dimensional analysis. Also, the primary dimensions of the stimulus that correlate with firing in these STC- TPM+ cells are well-described by the relevant subspace, as shown in Sup Fig 3g.

The biological insights that the method brings are:

- (1) Even if not all cells fire enough to give significant STC filters, we can use the general feature selectivity learned from the population of filter-producing cells to study more sparsely firing cells in the population.
- (2) The ‘snapshot’ that is considered for STC analysis does not capture the complete picture of the stimulus dependence. There are richer temporal dynamics that need to be considered.

Manuscript change summary:

To address all of these concerns, the changes in the manuscript are:

- (1) We added 5 panels to Supplementary Fig 3, and the firing rates presented in the table above are given in its caption.

Supp Fig 3 Caption:

(c) PS-STA population averages in wS2 for cells that gave significant filters from spike-triggered covariance analysis (blue curve, mean firing rate 2.9 Hz), cells that had significant PS-STAs but did not give STC filters (cyan curve, mean firing rate 1.6 Hz), and cells that had no significant response to either analysis method (orange curve, mean firing rate 0.8 Hz). The

STC threshold of 8 standard deviations above the shuffled distribution is shown on the graph to illustrate that cells that do not have significant STC filters have weaker PS-STA responses, but the responses are present. STC takes a snapshot only at time=0 on this graph and the TPM looks at the full temporal dynamics **(d)** The same as (c) but normalized to the maximum to highlight shape differences between the PS-STAs of wS2 cells that give STC filters and those that do not give STC filters **(e-f)** The same as (c-d) but for wS1 (mean firing rates are 2.0, 1.1, and 0.6 Hz for red, magenta and orange respectively) **(g)** Left: The eigenvector with the largest eigenvalue from cells with significant PS-STAs that did not give STC filters (cyan) in wS2 projected into the relevant filter subspace shown in Figure 1. Center: The same as the left but for cells that did not have significant PS-STAs or give STC filters. Right: Superposition of the two graphs to the left **(h-i)** Distribution of the negative peak time

(2) We have updated the main text by adding the following paragraphs:

wS2 has a distinct dynamic stimulus dependence from wS1 section

“The cells that have significant PS-STAs but do not show significant STC filters are still well-tuned to the relevant subspaces and exhibit temporal dynamics similar to the STC filter-producing cells (Supplementary Fig. 3c-g).”

TPM uncovers whisker-pad scale spatio-temporal dynamics

“These videos emphasize that important information is present beyond the static feature detection assumed by STC analysis. The stimulus dependence is not fully described within a small temporal window before a spike.”

Methods Section

“The TPM method goes beyond the traditional STC analysis because it can describe the behavior of neurons during continuous stimulation that do not produce STC filters. This is because it is focused on the empirically determined relevant dimensions (determined by a population analysis of filters), and thus avoids the problems associated with high dimensionality that are encountered with traditional STC. A neuron that does not have significant STC filters may still respond to stimuli with high projections into the empirically determined feature space. The ‘snapshot’ that is considered for STC analysis does not capture the complete picture of the stimulus dependence. There are richer temporal dynamics that need to be considered.”

REVIEWERS' COMMENTS:

Reviewer #2 (Remarks to the Author):

I thank the authors for their work. They have addressed my issues by clarifying the logic and content of their analyses. I think the manuscript -- which, before, was not clear enough in its description of the method's rationale -- has now improved substantially in its presentation.

The results and insights provided by the paper will be of interest to the community and I support publication.